# Contribution and functional connectivity between cerebrum and cerebellum on sub-lexical and lexical-semantic processing of verbs

**Azalea Reyes-Aguilar**[1]*, **Giovanna Licea-Haquet**[2], **Brenda I. Arce**[1], **Magda Giordano**[2]

**1** Department of Psychobiology and Neuroscience, Faculty of Psychology, Universidad Nacional Autónoma de México, Mexico City, Mexico, **2** Department of Behavioral and Cognitive Neurobiology, Instituto de Neurobiología, Universidad Nacional Autónoma de México, Queretaro, Mexico

\* azalea@neurocogcialab.org

**Data Availability Statement:** All data files are available from the OpenNeuro database, OpenNeuro Accession Number ds003481

## Abstract

Language comprehension involves both sub-lexical (e.g., phonological) and lexical-semantic processing. We conducted a task using functional magnetic resonance imaging (fMRI) to compare the processing of verbs in these two domains. Additionally, we examined the representation of concrete-motor and abstract-non-motor concepts by including two semantic categories of verbs: motor and mental. The findings indicate that sub-lexical processing during the reading of pseudo-verbs primarily involves the left dorsal stream of the perisylvian network, while lexical-semantic representation during the reading of verbs predominantly engages the ventral stream. According to the embodied or grounded cognition approach, modality-specific mechanisms (such as sensory-motor systems) and the well-established multimodal left perisylvian network contribute to the semantic representation of both concrete and abstract verbs. Our study identified the visual system as a preferential modality-specific system for abstract-mental verbs, which exhibited functional connectivity with the right crus I/lobule VI of the cerebellum. Taken together, these results confirm the dissociation between sub-lexical and lexical-semantic processing and provide neurobiological evidence of functional coupling between specific visual modality regions and the right cerebellum, forming a network that supports the semantic representation of abstract concepts. Further, the results shed light on the underlying mechanisms of semantic processing and contribute to our understanding of how the brain processes abstract concepts.

## Introduction

Language comprehension of spoken words requires sub-lexical (e.g., phonological), lexical-semantic [1–3], and syntactical processing [4]. According to the dual stream model, the neurobiological substrate in charge of these processes involves two networks in the left perisylvian brain regions [3, 5]. The first network is a dorsal stream, strongly left-dominant, integrated by structures in the posterior and ventral frontal lobe, posterior temporal lobe, and parietal

https://openneuro.org/datasets/ds003481/versions/1.0.3, and the DOI (https://doi.org/10.3389/fnhum.2021.666210) where the data description can be found in a publication.

**Funding:** MG: This study was supported by grants from DGAPA-PAPIIT IN203818 and CONACyT Fronteras de la Ciencia #225. The funders had no role in study design, data collection and analysis, decision to publish, or preparation of the manuscript.

**Competing interests:** The authors have declared that no competing interests exist.

operculum, involved in translating acoustic speech signals into articulatory representations, essential for speech development and normal speech production [3]. The second network, less understood than the former, is the ventral stream, which is organized bilaterally and consists of structures in the anterior and posterior temporal lobe involved in processing speech signals by mapping phonological representations onto lexical-semantic representations, or sound to meaning mapping [1].

According to Coltheart's dual-route model of visual word recognition [6], during reading there are two different routes: a direct or lexical route when recognizing full words based on their stored representations—requiring the perisylvian ventral stream mechanism [1, 7]—and an indirect or non-lexical route which involves the assembly of words from individual letters (i.e., graphemes) and sound (i.e., phoneme) as a function of the dorsal stream [1]. The non-lexical route is involved in reading new or unknown words or pseudo-words, i.e., pronounceable letter strings (e.g., RAME), for which there is no lexical-semantic representation. For pseudo-words, the perceptual analysis must be reconstructed letter-by-letter during listening or reading to build a word-form representation in phonetic languages such as Spanish. Previous studies have found that reading words involves a left language network: superior temporal gyrus/sulcus, middle temporal gyrus, angular gyrus, inferior frontal gyrus, and middle frontal gyrus, whereas pseudo-word reading produces activation in an attentional network that includes anterior/posterior cingulate, parietal cortex [8–10], and left ventral frontal regions [11]. Another study found that pseudo-words elicit more activation in the left dorsal stream than words, including the supramarginal gyrus in the temporoparietal cortex, precentral gyrus, and insula [12]. All of these brain regions, then, may be part of the non-lexical dorsal route considered in the dual-route model [6, 13].

Previous studies have shown that language processing involves left perisylvian regions and extends into sensory-motor and subcortical areas such as the thalamus and cerebellum [11, 14–17]. Motor brain systems demonstrably contribute to the comprehension of concrete and abstract concepts; their recruitment may be a flexible function of contextual semantic representations [18–21]. According to the modality-specific theory, these brain regions may be involved in semantic representation because said representation results from the rehearsal of sensory-motor experiences related to the specific meaning associated with each word [22]. The modal systems would then work together in the semantic processing of both concrete words and some abstract words with activation of consolidated sensory-motor experience [9, 18–21]. The embodied and grounded cognition approach offers an alternative: it proposes that conceptual representation is grounded in sensory-motor processes associated with specific contexts and situations [23].

On the other hand, so-called amodal theories claim that semantic processing recruits multi-modal and integrating brain regions given that sensory-motor formats become lost or detached during information integration [24]. For example, according to modality-specific theory, the semantic representation of verbs such as 'write' involves sensory-motor systems related to the action of moving and the motor experience of the most dexterous hand [20, 25]. Amodal theories propose instead that semantic integration mechanisms occur in multimodal regions such as the left perisylvian regions [26].

Although most studies on semantic representation have focused on nouns, verbs have become increasingly important in studying both semantic and syntactic processing. Noun processing has been associated with object knowledge representation in the left inferior temporal regions, while verb processing has been considered more complex, involving action knowledge, morphosyntactic processing, and executive functions related to the prefrontal and motor cortex [27–33]. There is demonstrated involvement of motor and pre-motor brain regions during the reading of verbs denoting motor actions [26, 34, 35]. However, when semantic

content is matched between nouns and verbs, such as comparing action nouns to action verbs, the differences between nouns and verbs become minimal or nonexistent, challenging psycho-linguistic models that support the organizational principle of grammatical class in lexical knowledge [36–38]. Verbs, as a fundamental word class, typically represent actions belonging to various semantic classes, including mental, relational, motor, verbal, existence, modulation, and states [39]. Meaning is multidimensional, encompassing various experiential components or aspects that can be encoded or evoked simultaneously, which allows for the possibility of a single verb being associated with distinct semantic classes [39]. For example, the verb "turn" can be associated with the motor semantic class, indicating the action of rotating or changing the direction of an object. It can also be linked to the state semantic class, indicating the process of becoming something or undergoing a change in nature or character. However, in this study, our specific focus is on two semantic classes of verbs: concrete-motor movement verbs, which encompass various subclasses of physical (non-mental) processes related to movements in space, posture-position, and modification of objects (e.g., 'write'), and abstract-mental action verbs (e.g., 'think') which were selected as they refer to entities endowed with psychic life that maintain or experience different states, changes of state, or inner perceptual, sensory, and/or cognitive activities [39–41]. In a functional magnetic resonance imaging (fMRI) study on Spanish verbs [42], concrete-motor movement verbs ("concrete" from here on) and abstract-mental action verbs ("abstract" from here on) were compared to pseudo-verbs using a region of interest (ROI) approach on frontal and temporal regions. The findings revealed that activation for concrete verbs occurred in frontal motor regions while activation for abstract verbs was observed in areas anterior to frontal motor regions. Similarly, Dalla Volta et al. [26] observed increased activation in the right somatosensory regions and the left parietal lobe for concrete verbs, whereas abstract verbs evoked higher activation in the prefrontal cortex outside the motor areas. These results support the hypothesis that abstract concepts recruit multi-modal brain regions, while concrete concepts activate sensory-motor regions.

Conversely, recent studies have demonstrated that abstract concepts also exhibit motor and sensory properties. The semantic representation of these concepts involves the brain's modal systems for perception and action related to direct or simulated sensorimotor experience that contribute to abstract meaning [43], which supports the embodied and grounded cognition models [18, 19, 21]. The extent of involvement of modal systems in the semantic representation of abstract verbs may vary depending on contextual factors or strategies, such as the utilization of visual experiences and imagery [9, 23]. Sensory and motor information has been found to play a significant role in the semantic content of abstract nouns [9, 18–21]. Harpaintner et al., (2020) found that abstract concepts related to action and vision are grounded in modality-specific brain systems typically engaged in actual perception and action. In the case of verbs, another study [44] identified examples of abstract verbs that elicit similar outcomes to visual imagery when they are straightforwardly interpreted as referring to spatially concrete scenes. In this regard, visual brain regions have been shown to have a significant role in processing certain abstract verbs (mental verbs such as 'understand') that are associated with visual information, such as "see" or "look" [19, 22]. Vision often plays a dominant role in our interpretation of sensory information [45] and extends to the semantics of abstract concepts, with hearing a close second [46]. Likewise, and more consistently, cortical and subcortical motor regions are involved in semantic representations of motor movements, e.g., motor verbs such as 'write' [19, 22], as well as regions related to lateralized dominance for behavioral motor proficiency, motor experience, and handedness [47, 48]. In sum, there is still controversy regarding the differential recruitment of brain regions between concrete and abstract verbs.

Despite the recent increase in the number of studies about verbs, the neural basis of their sub-lexical and lexical-semantic processing warrants further inquiry. In this study, we focused on verbs as a lexical category that can represent concrete or abstract actions. To compare the contribution and functional connectivity of brain regions in sub-lexical and lexical-semantic processing of verbs, we designed stimuli that encompassed pseudo-verbs, strings of letters featuring characteristic verb infinitive endings in Spanish (i.e., ar, er, ir), and actual verbs. While reading or identifying stimuli engage similar visual processing flows regardless of stimulus type, only verbs and pseudo-verbs require phonological processing; similarly, only verbs involve lexical-semantic information [4, 49, 50]. We consequently contrasted verbs, which we expected would recruit the ventral stream, with pseudo-verbs, which would activate the dorsal stream. Likewise, we were interested in the dichotomous brain representation of concrete versus abstract verbs because, as mentioned above, the brain correlates are controversial [18, 23, 26, 51]. Thus, we were evaluating the involvement of integrating multimodal and modality-specific brain regions in the processing of verb categories. According to the embodied or grounded cognition approach, we would expect to find that both multimodal and modality-specific cortical regions would be involved in processing both categories [18, 52]. However, we expected a variation in involvement of modality-specific regions between verb categories: motor verbs would primarily engage motor brain areas, while mental verbs would activate sensory brain areas, with a dominant focus on visual processing, as observed in previous studies [44, 46].

## Methods

### Participants

Twenty-four volunteers (12 women) between 21 and 35 years old (M = 26.75, SD = 3.95), all native Spanish speakers, participated in this study. Participants were recruited between October 2017 and March 2018. Only two authors involved in the collection of data had access to individual participant data during that research phase; afterwards, the data was anonymized. Sample size was calculated to detect a within-subject effect of a large size ($d = 0.8$), with 80% power, and a low alpha value (0.01) using a two-tailed one-sample $t$-test. Participants showed no neurological or psychiatric disorders according to the Mexican version of Symptom Checklist 90 [53]. After the participants were informed of the study procedures and confidentiality, they signed the written informed consent to participate in the experiment. The experimental protocol was approved by the local Ethics in Research Committee of the Instituto de Neurobiología-UNAM (#047.H.RM) in compliance with the Mexican Health Department's federal guidelines, which adhere to international regulations.

### Stimuli

We selected Spanish verbs according to two categories, concrete-motor and abstract-mental, based on psycholinguistic properties from the ADESSE database (http://adesse.uvigo.es/). For motor verbs, we selected verbs from the material class related to movements in space and modification of objects; for mental verbs, we selected those that belong to the mental class associated with perception, cognition, and choice (*elección* in Spanish), generating a list of 330 verbs: 167 motor verbs and 163 mental verbs. We then obtained the use frequency of each verb through Sketch-Engine ([54] http://www.sketchengine.eu) and LEXMEX-Spanish [55] corpora. If a verb was not in both corpora, it was excluded from the study (seven verbs were excluded). The verbs were then arranged by use frequency and matched across category based on their number of syllables; two were excluded because of length (i.e., they had four or more syllables). The final selection included 288 verbs: 139 mental verbs and 149 motor verbs.

These 288 verbs were presented to an independent sample from that of the fMRI study, consisting of thirty young participants (18 women) aged between 20 and 35 years (M = 24.11, SD = 4.57), all of whom were native Spanish speakers. They were asked to read and rate the verbs on five psycholinguistic properties using a 1–6 Likert scale: motor-mental relatedness, concreteness-abstraction, imageability, emotional valence, and arousal. According to a Wilcoxon Signed-Rank test, there were differences between both categories in four out of five psycholinguistic properties. Motor verbs were classified as motor and mental verbs as mental (Z = 162, p < 0.001). In the concreteness-abstraction scale, motor verbs obtained higher concreteness scores while mental verbs were classified as more abstract (Z = 247, p < 0.001); motor verbs were rated with higher imageability than mental verbs (Z = 18340, p < 0.001); and mental verbs showed higher scores in arousal compared to motor verbs (Z = 8323, p < 0.01). There was no discernible variation in emotional valence between verb categories. We selected the motor verbs with a higher perceived motor relatedness (i.e., scores < 3), and mental verbs with higher mental relatedness (i.e., scores > 4), with no overlapping scores between categories. Similarly, there was no rating overlap across verbs in one category over another on concreteness-abstraction (motor verbs had scores < 3, and mental verbs, > 3) nor in imageability (motor verbs scored = 6, and mental verbs < 6). As a result, we were able to generate two verb category lists with entries with similar use frequency and number of syllables, but different degrees of motor-mental relatedness, concreteness-abstraction, and imageability (i.e., 112 motor-verbs, e.g., *caminar* [walk], and 112 mental-verbs, e.g., *entender* [understand]). We then created pseudo-verbs with a comparable level of pre-lexical familiarity. For each verb, a pseudo-word was created, changing the position of consonants but maintaining the vowels and the lexical indicator for verb endings in Spanish [ar, er, ir], For example, for the verb *caminar* [to walk], the pseudo-verb *nacimar* was generated, which conserves the phonological (i.e., sub-syllabic) structure of the word. All resulting letter sequences were pronounceable and were run through an authoritative Spanish dictionary (*Real Academía Española*, https://dle.rae.es/nacimar?m=form), to eliminate any letter sequences with an entry. 112 pseudo-verbs were then randomly selected for the fMRI study. One final stimulus category was included. A string of symbols was generated for each verb, in which a pair of symbols or characters (e.g. %%) stood in for each word syllable (e.g., for "*caminar*" which has three syllables, the symbol stream was $$##&&, or three pairs of symbols). Half of these symbol strings were then randomly selected to obtain 112 stimuli.

## Identification task

We included four stimulus categories: mental verbs, motor verbs, pseudo-verbs, and symbols. The stimuli were presented using a block design, each block consisting of stimuli from the same category. There were seven different stimuli in each block, three of which were repeated consecutively, resulting in a total of ten stimuli per block. The stimuli were displayed for 1000 ms with interstimulus intervals of 2000 ms. Blank intervals of 12 seconds separated the blocks (Fig 1). Two blocks of each stimulus category were presented pseudo-randomly during each run. The total number of runs was eight. Each run lasted approximately 6 minutes. Participants were instructed to identify the current stimulus (i.e., verbs, pseudo-verbs, and symbols strings) and indicate during the stimulus presentation whether it was the same as the previous one (one-back detection task). The stimuli were presented on a gray background using the PsychoPy® software [56, 57], and a projection system consisting of MR-compatible goggles from NordicNeuroLab (Bergen, Norway). Additionally, a button system from NordicNeuroLab was used to record the participants' responses.

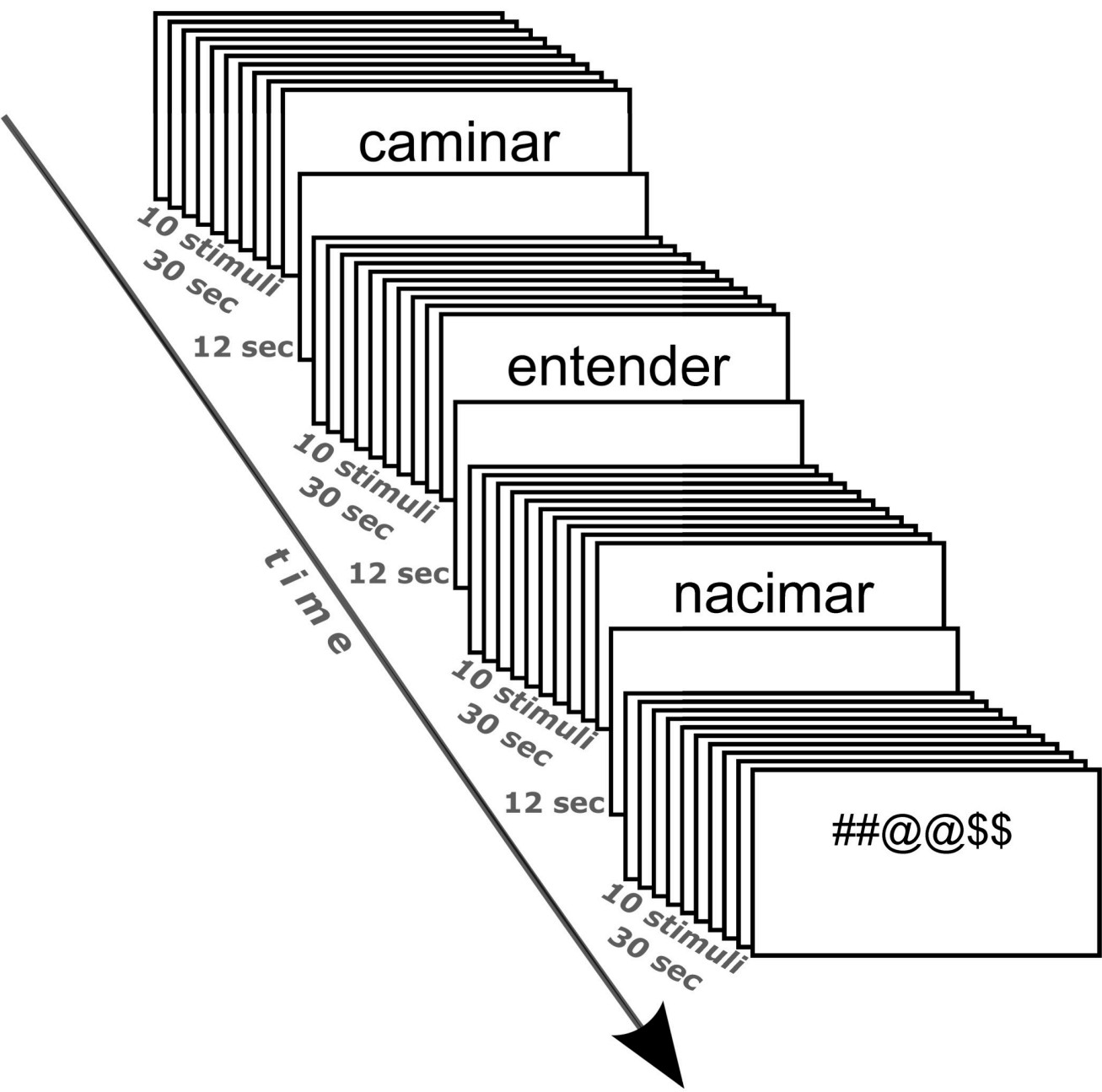

**Fig 1. The stimuli were presented for 1000 ms with 2000 ms inter-stimulus intervals.** Each block included ten stimuli from one category, and subjects performed a one-back detection task;12-sec blank intervals separated blocks. Two blocks of each stimulus category were presented pseudo-randomly in each of the eight runs.

## Procedure

**fMRI study.** As mentioned earlier, the scanning procedure involved eight runs in which participants were instructed to identify the current stimulus (i.e., verbs, pseudo-verbs, and symbols strings) and indicate during the stimulus presentation whether it was the same as the previous one (one-back detection task).

**Behavioral testing.**    On a different day, participants returned to complete the vocabulary subscale of the Wechsler Adult Intelligence Scale (WAIS-IV [58]) as a measure of lexical performance, and three varieties of verbal fluency tasks. For these tasks, participants were instructed to generate words for one minute in three different categories: verbs as a lexical category, animal names (semantic fluency), and words starting with the letter "m" (phonological fluency). The letter "m" in Spanish has a frequency of use of 3.4% (Diccionario del Español de México: http://dem.colmex.mx), making the task moderately challenging for the participants and within the range of use frequency (from 1 to 10%) of letters selected in previous studies [e.g., 59–61]. To avoid a fatigue effect, the order of behavioral tasks was counterbalanced at the subject level. Finally, the Edinburgh Handedness Inventory (EHI [40]) was administered to assess hand dominance in everyday motor activities. The EHI index, which ranges from -100 to 100, was calculated. This index enables the classification of participants with negative scores from -40 to -100 as left-handed, scores between -40 and 40 as ambidextrous, and scores greater than 40 as right-handed.

**Imaging data acquisition.**    fMRI imaging was performed on a 3.0T GE MR750 scanner (General Electric, Waukesha, WI) using a 32-channel head coil. Functional imaging included 38 slices, acquired using a T2*-weighted echo-planar imaging sequence with TR/TE 2000/40 ms, field of view of 25.6 cm, a $64 \times 64$ matrix, and 4-mm slice thickness, resulting in a $4 \times 4 \times 4$ mm3 isometric voxel. High-resolution structural 3D-T1-weighted images were acquired for anatomical localization. These images were acquired with 3D spoiled gradient recall (SPGR), resolution of $1 \times 1 \times 1$ mm3, TR = 8.1 ms, TE = 3.2 ms, flip angle = 12˚, inversion time = 0.45, covering the whole brain.

**fMRI data analysis.**    For quality control of the BOLD data, following a maximum absolute motion of more than 2mm as an exclusion rule, one of eight runs was excluded for three subjects. MRI data were analyzed using FSL (FMRIB's Software Library, www.fmrib.ox.ac.uk/fsl [62]). Statistical analysis was performed with FMRI Expert Analysis Tool using FMRIB's Improved Linear Model (FEAT FILM) Version 6.0.1. Each participant's data were brain extracted, motion and slice timing corrected, and normalized onto MNI common brain space (Montreal Neurological Institute, EPI Template, voxel size 2mm x 2mm x 2mm). Data were smoothed using a gaussian filter (full-width half maximum = 6mm) and high-pass filtered during analysis. Blood oxygen level-dependent (BOLD) signal was examined during the stimuli presentation when participants were instructed to read or identify the stimulus (verb, pseudo-verb, or symbols). Statistical analysis of event-related hemodynamic changes was carried out per the general linear model (GLM [63]). The model included the following regressors: motor verbs, mental verbs, pseudo-verbs, and symbols. The regressors were created to specifically represent the duration of stimulus presentation (1000 ms) for each set of ten stimuli within the block, resembling an event-related design. First-level fMRI analysis data was performed to identify regions that increased BOLD signal intensity for each of the four categories of stimuli relative to blocks of blank intervals for each run with a significance threshold criterion of $Z > 2.3$. Since each subject responded to the experimental paradigm in eight independent runs, to estimate a map of the brain regions involved during the identification of each stimuli category, a mid-level analysis was carried out using a fixed-effects model, which averaged the activity of verbs and pseudo-verbs during each of the eight runs respect to symbols: a conjunction analysis was done with both verbs categories, motor-verbs ∩ mental-verbs > symbols contrast, and pseudo-verbs > symbols contrasts. To test whether verbs recruited different mechanisms relative to pseudo-verbs, a conjunction analysis was done with both verb categories: motor-verbs ∩ mental-verbs > pseudo-verbs, and pseudo-verbs > motor-verbs ∩ mental-verbs contrasts. Then, we compared verb categories: motor > mental and mental > motor contrasts. Finally, we conducted this analysis with the removal of mechanisms related to visual

processing (symbols) and phonological processing (pseudo-verbs): [motor > symbols] > [mental > symbols], [mental > symbols] > [motor > symbols], [motor > pseudo] > [mental > pseudo], and [mental > pseudo] > [motor > pseudo] contrasts. To identify activations at the group level, we used a third-level analysis using FLAME 1 (FMRIB's Local Analysis of Mixed Effects) with a cluster significance threshold criterion of $Z > 2.3$ with $p < 0.05$ corrected for multiple comparisons with Gaussian Random Field (GRF) for results at the whole-brain level [64].

**Region of interest (ROI) analysis.** ROI analysis was performed on ten regions detected as maxima from an automated meta-analysis using "verbs" as a term in neurosynth (http://neurosynth.org/ [65]). For the voxel as a maxima activation, an 8-mm *spherical* ROI was built for each ROI. Nine ROIs were localized on the cerebral cortex (seven on the left hemisphere, two on the right hemisphere according to Harvard-Oxford Cortical Structural Atlas), and the last one, on the right cerebellum (Cerebellar Atlas in MNI152 space). The ROIs were located as follows: left temporal occipital fusiform cortex (L-FusOcc [-40, -50, -20]), left inferior frontal gyrus (L-IFG [-50, 16, 16], pars opercularis), left superior lateral occipital cortex (L-LOC [-26, -60, 48], superior parietal lobule and angular gyrus), left anterior and posterior middle temporal gyrus (L-MTG-ant [-60, -50, 0], and L-MTG [-58, -10, -14], respectively), left supplementary motor area (L-SMA [-2, 8, 60]), left supramarginal gyrus (L-SMG [-60, -26, 28]), right orbitofrontal cortex (R-orb [36, 26, -4]), right superior temporal gyrus (R-STG [56, -32, 2]), and right cerebellum (R-Cerebellum [34, -64. -29], crus 1 and lobule VI). Then, for each ROI, the average percent signal change from calculated semantic contrasts in the GLM (mental vs. motor verbs) was calculated. Finally, a correlation analysis was carried out with these values and the scores obtained from the behavioral tests.

**Psychophysiological interaction analysis.** To test the functional connectivity modulated by phonological and lexical-semantic processing, psychophysiological (PPI) analysis was conducted with those seeds ROIs that showed a correlation between semantic contrast, i.e. mental > motor, in the GLM analysis and scores from behavioral tests (i.e., L-SMG, L-LOC, L-MTG, L-SMA, and R-Cerebellum). In addition, we conducted a whole-brain PPI analysis for each seed to know how each one of the seeds (time series of the brain region, i.e., seed, as physiological regressor) was coupled with other brain regions during the reading of verbs as compared to symbols and pseudo-verbs (psychological regressors).

Each ROI was projected on the pre-processed functional images, i.e., eight runs for each participant. Then, the time series of BOLD activity was extracted using fslmeans utility as an average across all voxels within each seed ROI for each data set. Finally, the PPI analysis was conducted for every ROI separately using FEAT (FMRI Expert Analysis Tool) Version 6.00, part of FSL (FMRIB's Software Library www.fmrib.ox.ac.uk/fsl).

First-level GLM analysis included nine regressors: a physiological regressor (i.e., time series of the seed), four psychological regressors corresponding to the four stimulus categories (based on the design of the one-back detection task), and the last four regressors for interaction between psychological and each physiological regressor (PPI). Psychophysiological interaction was determined by testing for a positive slope of the PPI regressor. Individual contrast images for PPI analysis were the same as whole brain analysis and were entered into the subject-level analysis. Data for PPI analysis, as well as GLM, were processed with FEAT, part of FSL. Subject-level analyses were performed separately for each condition. Finally, the contrast images generated at the subject-level analyses were entered into the group-level analyses.

**Statistical analysis of behavioral measures.** Behavioral data were analyzed using R 3.6.1. We compared the performance among three verbal fluency tasks using a repeated-measures ANOVA, and calculated Pearson's correlation coefficient to test whether the scores of the

behavioral tasks were correlated. We also set a false discovery rate (FDR) correction to determine the threshold for significant differences (p<0.05).

## Results

### Behavioral measures—Linguistic performance

We observed a higher fluency of verbs (M = 24.50, SD = 6.87, range = 8–36, $F_{(2, 46)}$ = 33.66, $\eta2$ = 0.37, p < 0.0000000218, FDR corrected) and semantic fluency (M = 23.95, SD = 4.08, range = 15–33, p < 0.000000398, FDR corrected) relative to phonological fluency (M = 16.17, SD = 3.86, range = 8–26). No significant difference was found between semantic and verb fluency. We then examined the relationship between measures of verbal fluency: only verb fluency demonstrated a positive correlation with phonological fluency (r = 0.55, p = 0.014, Pearson, FDR corrected). Next, we evaluated the relationship between verbal fluency and the vocabulary task (M = 30.67, SD = 8.78, range = 8–43). The vocabulary task showed a positive correlation with both verb fluency (r = 0.60, p = 0.006, Pearson, FDR corrected) and semantic fluency (r = 0.54, p = 0.009, Pearson, FDR corrected, Fig 2), but we observed no significant correlation with phonological fluency. According to the EHI scores (M = 76.04, SD = 21.37, range = 10–100), which serve as a measure of everyday motor behavior laterality, 8.84% of participants were classified as ambidextrous (score < 40), while 91.67% were right-handed (score > = 40). No left-handed individuals were included in the study. Finally, we explored the relationship between EHI scores and performance on the language tasks: none of the results remained statistically significant after applying multiple comparison correction (FDR).

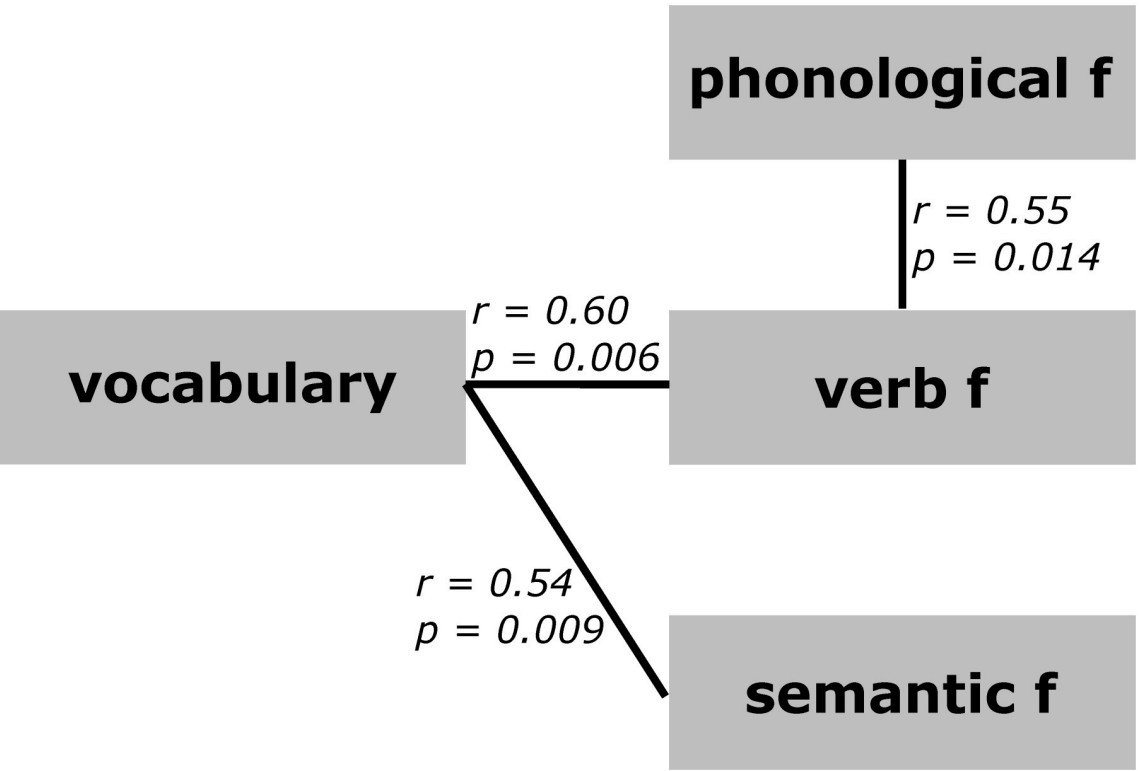

**Fig 2. Correlations among scores on behavioral tasks.** Vocabulary task of WAIS-IV, verb, semantic, and phonological fluency (f).

## fMRI results

In the fMRI study, the participants successfully performed the one-back detection task with an overall 99% success rate, which confirmed that the participants did pay attention to the stimuli for visual, phonological, and lexical processing to determine whether two consecutive stimuli were the same or not. There were no differences in correct response rate across stimuli categories -i.e. mental verbs, motor verbs, pseudo-verbs, and symbols- therefore any variation in the BOLD signal cannot be due to the difficulty of the stimulus categories.

Results from the GLM analysis showed that, compared to symbols, verbs recruited bilateral frontal, temporal, and parietal cortex regions (Fig 3, S1 Table). In contrast, pseudo-verbs showed involvement of the left hemisphere language network extending into motor areas such as pre and postcentral gyrus, SMA, and right cerebellum (crus I, crus II, lobules VI, and VIIIa; Fig 3 and, S1 Table).

Verbs, in contrast to pseudo-verbs, recruited left posterior temporal, right middle temporal, ventral parietal, and frontal regions with more extension on the right hemisphere, in addition to medial regions: anterior and posterior cingulate cortex and occipital areas, i.e., lingual gyrus (Fig 4, S2 Table). Comparing in the opposite sense, pseudo-verbs recruited more left-lateralized frontal motor regions, pre and postcentral gyrus, SMA, and right cerebellum (crus I, crus II, lobules VI and VIIb), bilateral superior lateral occipital cortex, and left temporal occipital fusiform cortex including the visual word form area as compared to verbs (Fig 4, S2 Table).

When comparing the effect of semantic differences across verb categories, we found no effect for the motor > mental contrast. Conversely, the mental > motor contrast evidenced an increase of BOLD signal on the left inferior frontal region, dorsomedial prefrontal cortex, including SMA, left ventral temporal areas and medial occipital regions, and right cerebellum (crus II and lobule VI, Fig 5, S3 Table). Removing the effect of visual processing (i.e., symbols) in the comparison [motor > symbols] > [mental > symbols] resulted in the recruitment of bilateral lateral superior occipital cortex and ventrolateral temporal regions, including the

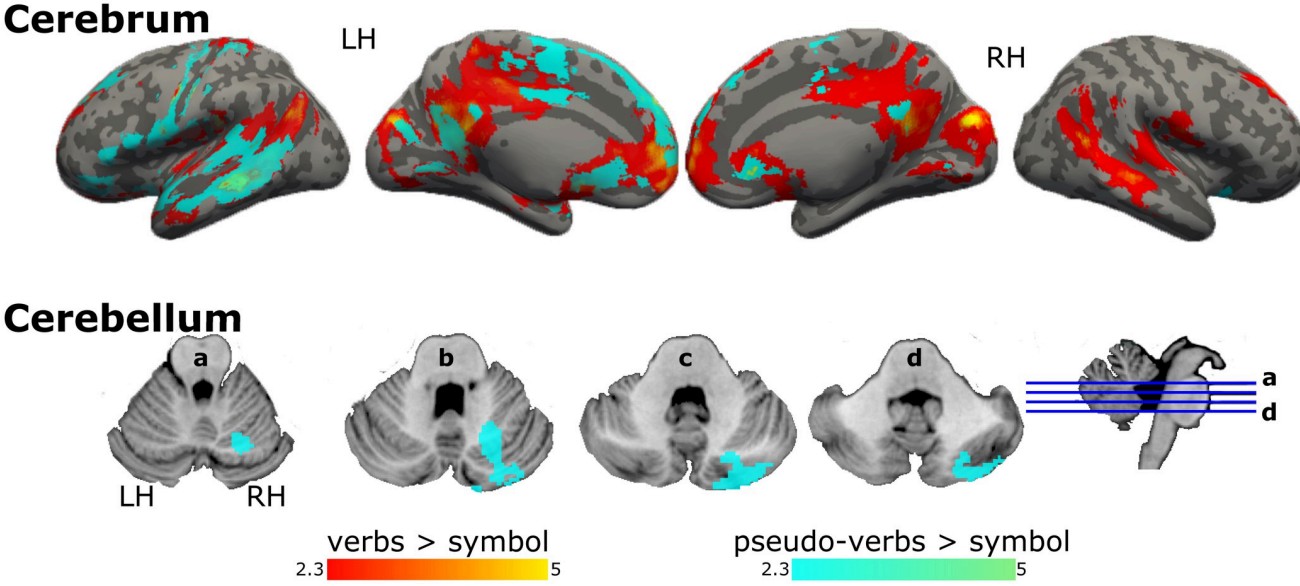

**Fig 3. Activation maps for verbs and pseudo-verbs with respect to symbols.** Graphical representation of GLM's results, brain regions activated in the contrast verbs > symbols in red, and pseudo-verbs > symbols in aqua. Coordinate z of multislices in cerebellum: a = -24, b = -29, c = -34, and d = -39. Color bars show z scores. LH: left hemisphere; RH: right hemisphere.

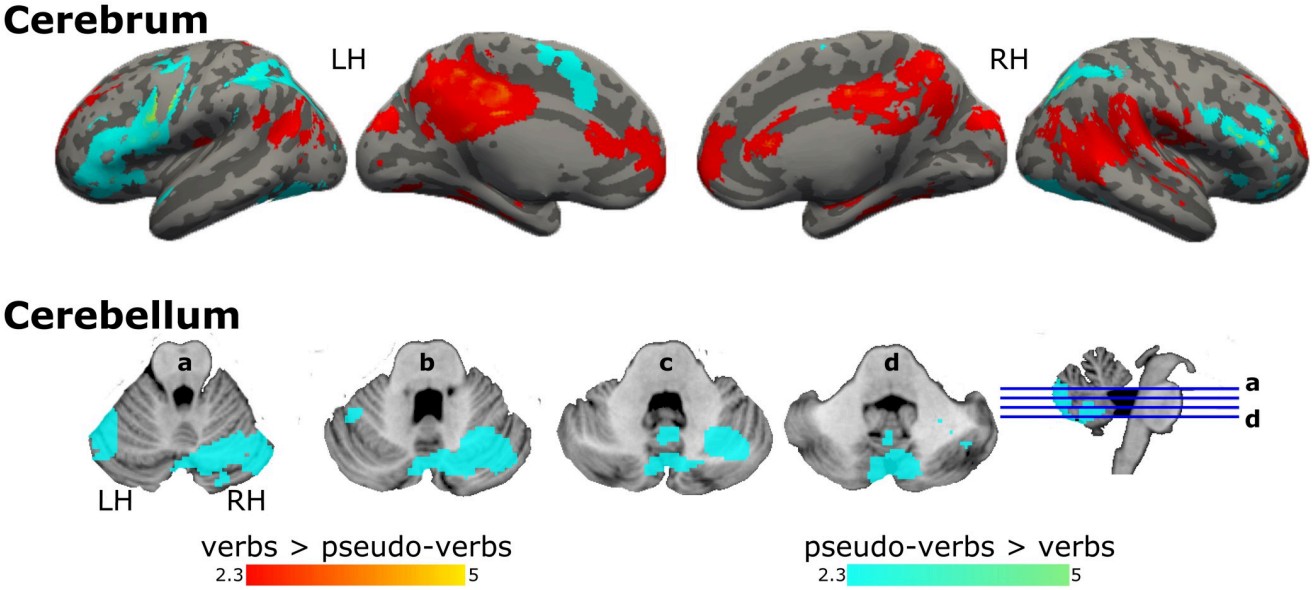

**Fig 4. Activation maps for the comparison between verbs and pseudo-verbs.** Graphical representation of GLM's results, brain regions activated in the contrast verbs > pseudo-verbs in red, and pseudo-verbs > verbs in aqua. Coordinate z of multislices in cerebellum:: a = -24, b = -29, c = -34, and d = -39. Color bars show z scores. LH: left hemisphere; RH: right hemisphere.

fusiform area (S1 Fig, S4 Table). Whereas the opposite contrast, [mental > symbols] > [motor > symbols] showed an increase of BOLD signal in medial frontal and occipital regions (Fig 5, S3 Table). Finally, when the phonological effect was removed (i.e., pseudo-verbs), the contrast [motor > pseudo] > [mental > pseudo] did not show significant differences in BOLD signal, however the opposite contrast, [mental > pseudo] > [motor > pseudo], recruited the medial occipital areas, and lingual gyrus (Fig 5, S3 Table).

## Region of interest (ROI) analysis

We analyzed the relation between behavioral tests and the BOLD signal changes observed in specific language brain areas during the identification task. We defined ten ROIs from a meta-analysis for "verbs" (see details in Methods).

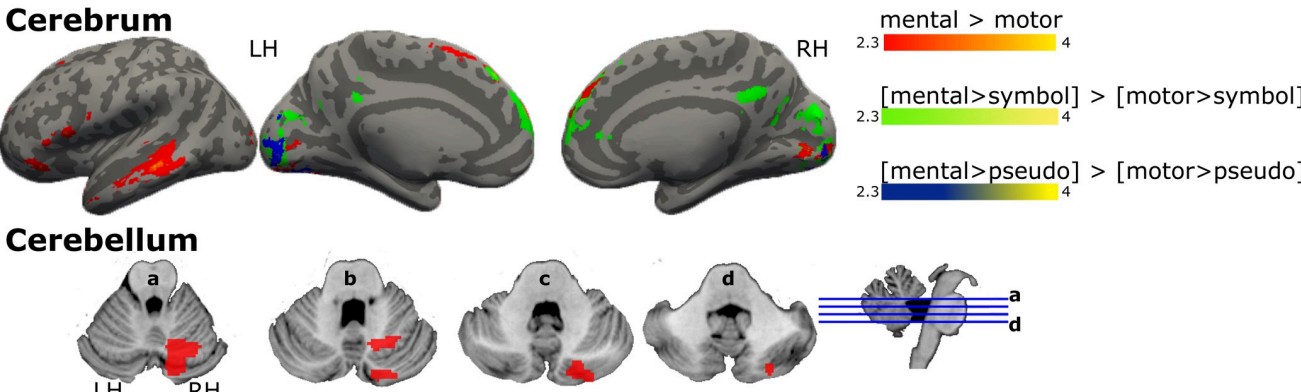

**Fig 5. Activation maps for the comparison between mental and motor verbs.** Graphical representation of GLM's results, brain regions activated in the contrast mental > motor verbs in red, same contrast when the effect of visual processing (i.e., symbols) is removed, in green, and when the effect of phonological processing (i.e., pseudo-verbs) is removed, in blue. Coordinate z of multislices in cerebellum: a = : a = -24, b = -29, c = -34, and d = -39. Color bars show z scores. LH: left hemisphere; RH: right hemisphere.

The BOLD signal change for mental > motor verbs contrast in the L-LOC ROI was positively correlated to verbs fluency (r = 0.53, p = 0.034, Pearson, FDR corrected) and negatively correlated to EHI (r = -0.50, p = 0.034, Pearson, FDR corrected). The L-MTG ROI was negatively correlated to EHI (r = -0.54, p = 0.030, Pearson, FDR corrected). Other significant correlations were detected between behavioral measures and BOLD signal change for other contrasts: verbs > symbols and verbs > pseudo-verbs (S5 Table). Since a significant correlation was observed between the BOLD signal in the mental > motor contrast with verbs fluency and EHI, we conducted an exploratory analysis to calculate partial correlations between EHI and BOLD signal, with verbs fluency as the covariate. For this, negative partial correlations were found for L-MTG (r = -0.58, p = 0.003, Pearson), both motor areas, L-SMA (r = -0.45, p = 0.029, Pearson), and R-cerebellum (r = -0.42, p = 0.045, Pearson). Therefore, we identified these four ROIs—L-LOC and L-MTG as semantic regions, and L-SMA and R-Cerebellum as motor areas—as having a significant relationship with verb fluency or EHI, making them suitable candidates for the analysis of functional connectivity.

## Psychophysiological interaction

Having established that those four ROIs (i.e., L-LOC, L-MTG, L-SMA, and R-Cerebellum) showed a relation to behavioral tasks in the contrast mental > motor verbs, suggesting semantic differences, we conducted a PPI analysis with these four ROIs as seeds. We analyzed functional connectivity of verbs relative to other categories, i.e., symbols and pseudo-verbs, and between categories of verbs, i.e., mental and motor. PPI analysis was calculated for the following contrasts: motor > symbols, mental > symbols, pseudo > symbol, motor > pseudo, mental > pseudo, motor > mental, and mental > motor.

First, we obtained the psychophysiological interaction for motor and mental verbs and pseudo-verbs, with respect to symbols. L-LOC showed functional connectivity with bilateral temporal and parietal regions for motor and mental verbs, while for pseudo-verbs the functional connectivity was only with left regions such as IFG, pre, and postcentral gyrus. L-MTG, as seed, showed functional connectivity with the right dorsolateral prefrontal cortex only for pseudo-verbs. L-SMA and R-Cerebellum were coupled only for mental verbs. L-SMA was functionally connected with bilateral frontal regions and thalamus, while R-Cerebellum with medial occipital regions, bilateral temporal regions, pre and postcentral gyri, and polar frontal regions (Fig 6, S6 Table).

Then, psychophysiological interaction was calculated for each category of verbs with respect to pseudo-verbs (Fig 7, S7 Table). For motor verbs, L-LOC was connected with bilateral parietal and temporal regions, and for mental verbs, with more extension in postcentral areas. L-SMA, for motor verbs, was connected with bilateral superior temporal regions, posterior cingulate cortex, and frontal pole; for mental verbs, it was connected with posterior cingulate cortex, medial prefrontal cortex, left dorsal parietal regions, and bilateral posterior temporal areas. R-Cerebellum was connected with other regions only for mental verbs: bilateral temporal regions, post and precentral gyri, frontal areas, and precuneus.

Finally, psychophysiological interaction was calculated between categories of verbs. The motor > mental contrast did not show significant interaction, while the mental > motor contrast evidenced an interaction between the right cerebellum and medial occipital area (Fig 8, S8 Table). The occipital area corresponds to 8.2% of the medial occipital regions that showed an increase in BOLD signal in the mental > motor activation map resulting from the GLM analysis (Fig 5).

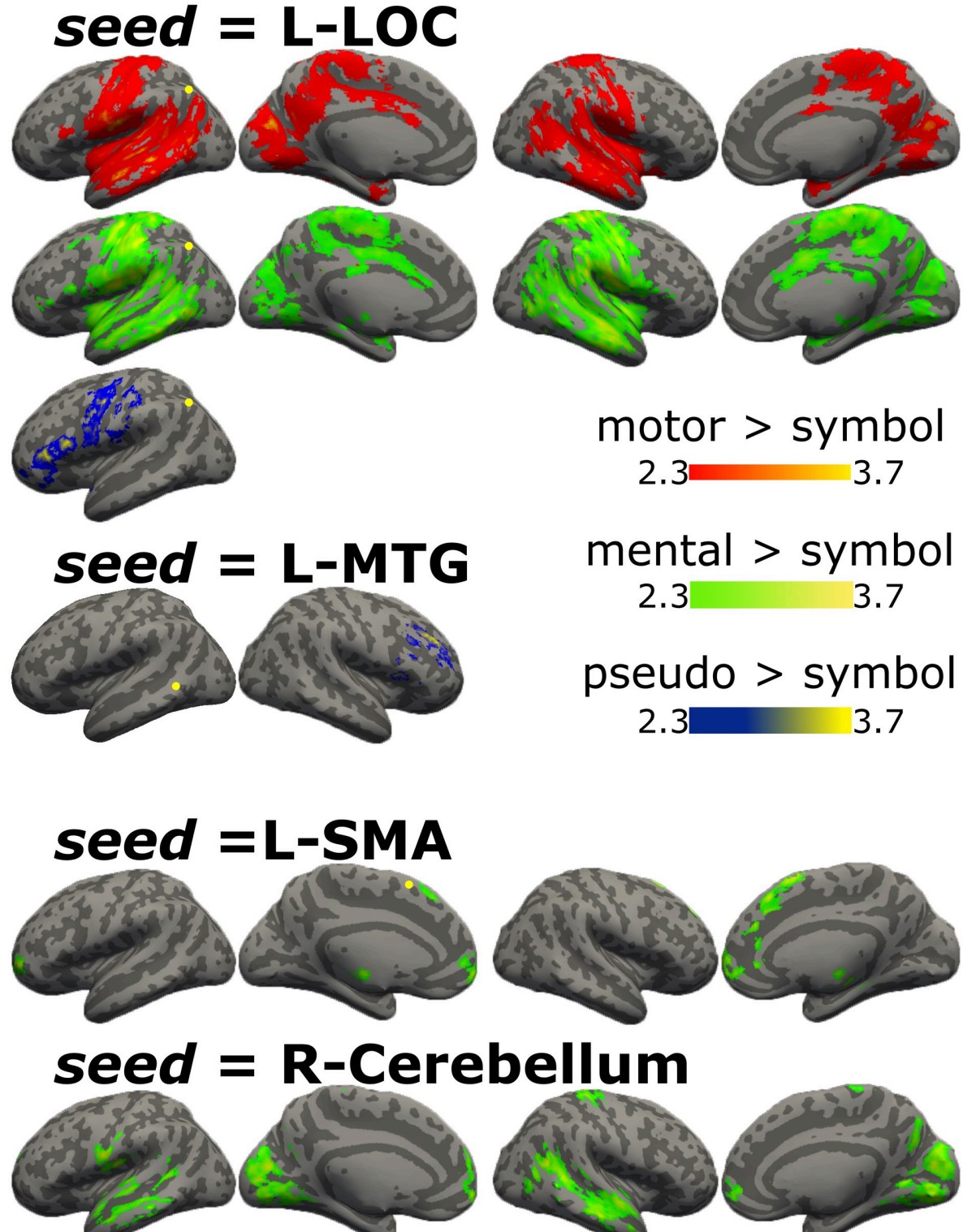

**Fig 6. PPI results for motor, mental and pseudo verbs with respect to symbols.** Graphical representation of PPI results with seeds in L-LOC, L-MTG, L-SMA, and R-cerebellum, brain regions functionally connected in the contrast motor > symbols in red, mental > symbols in green, and pseudo-verbs > symbols in blue. Color bars show z scores. LH: left hemisphere; RH: right hemisphere.

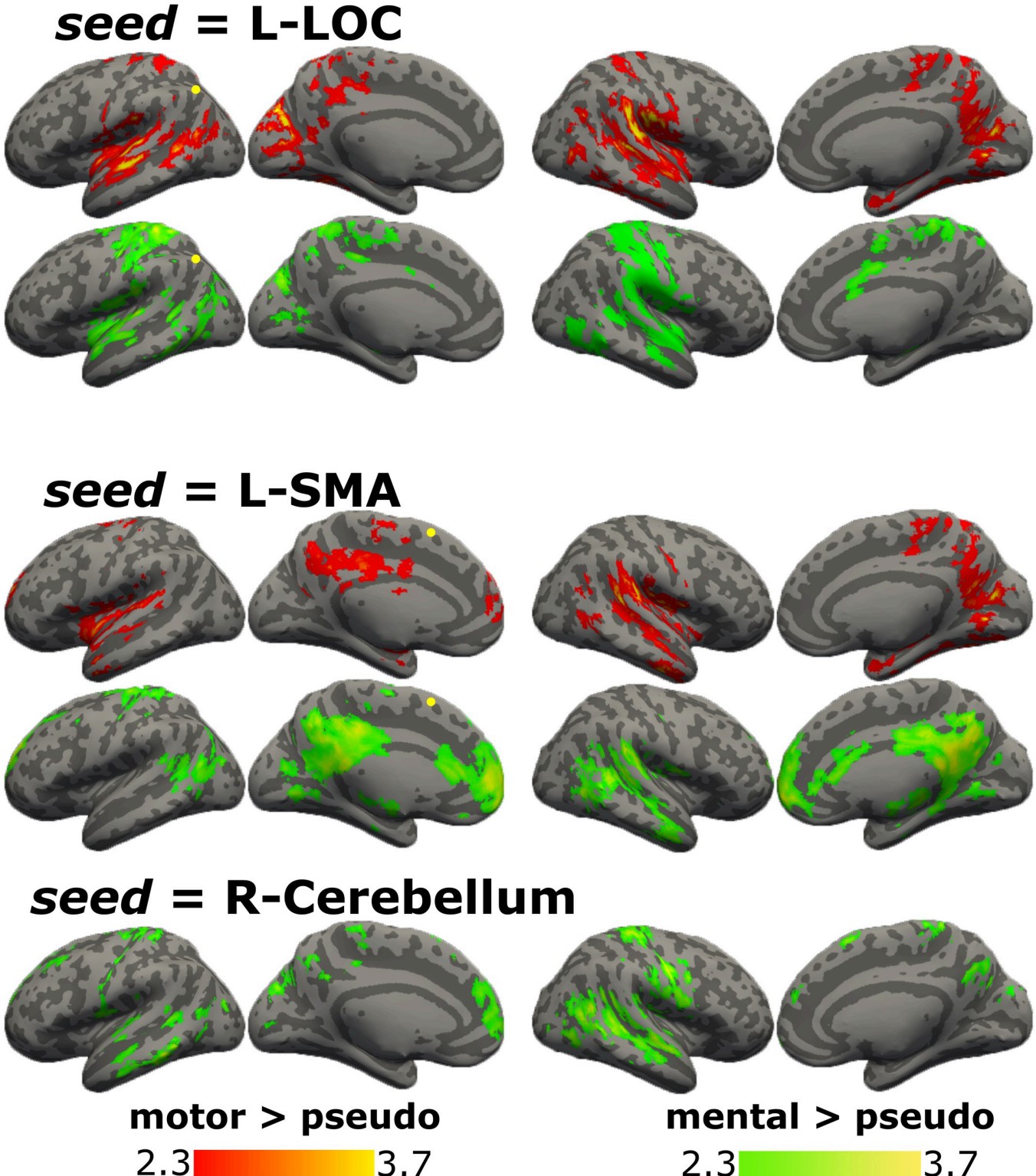

**Fig 7. PPI results for motor and mental verbs with respect to pseudo-verbs.** Graphical representation of PPI results with seeds in L-LOC, L-SMA, and R-cerebellum, brain regions functionally connected in the contrast motor > pseudo-verbs in red and mental > pseudo-verbs in green. Color bars show z scores. LH: left hemisphere; RH: right hemisphere.

# *Seed* = Cerebellum

**Fig 8. PPI results for the comparison between mental and motor verbs.** Graphical representation of PPI results with the R-cerebellum seed, brain regions functionally connected in the contrast mental > motor green. Color bar shows z scores. Coordinate z slice in cerebellum = -29. LH: left hemisphere; RH: right hemisphere.

## Discussion

This study confirmed the differential recruitment and functional connectivity of neural regions according to the sub-lexical processing level for pseudo-verbs and lexical-semantic level for verbs. Reading verbs involved bilateral frontal, temporal, and parietal regions as already described for reading neural networks [11]. Pseudo-verbs, which represented a sub-lexical level of processing, recruited the left language network and the extrasylvian motor system, including the right cerebellum. Differences in neural activation between semantic categories revealed that mental verbs, relative to motor verbs, exhibited greater activation of the left-language perisylvian network and extrasylvian regions such as the cerebellum and occipital areas, which showed functional connectivity. According to our expectation, these results showed a dominant involvement of visual modality-specific brain areas in processing mental verbs that convey abstract, non-motor actions, for example, "to understand".

The results of the behavioral tests showed better linguistic performance for verbs and semantic fluency than phonological fluency, consistent with previous findings among Spanish speakers and speakers of other languages [59–61, 66, 67]. We take this to mean that the organization of the mental lexicon favors lexical-semantic representations over phonological characteristics. Using the ROI approach, the contrast mental versus motor verbs detected that verbal

fluency was positively correlated with the change in BOLD signal in the left lateral occipital cortex. This region is considered a semantic region, as it receives inputs from higher-order representations to distinguish and integrate meaning [14, 68] for actions verbs and observed actions [9]. On the other hand, in this same contrast, lateralized motor behavior as measured by scores in the EHI was associated with less signal change in semantic areas, i.e., left middle temporal gyrus and left lateral occipital cortex.

In keeping with our expectations, verbs—in contrast to symbols and pseudo-verbs—recruited an extended reading network on bilateral temporal regions that belong to the ventral stream, related to semantic processing [1, 3] and medial regions related to attention [8, 68]. On the other hand, pseudo-verbs compared to verbs recruited prefrontal regions, perhaps related to Hagoort's [10] proposition that language processing recruits various prefrontal cortex regions: the ventral region integrates elements from memory to create novel meanings or interpretations, while the dorsolateral region, associated with attentional control, interacts with the language network to select the most relevant information for language comprehension. In the present study, the reading of pseudo-verbs recruited the prefrontal cortex possibly in a search for meaning for these unknown linguistic stimuli. In contrast, in the case of verbs, it seems there would be no need for the prefrontal regions to be selectively activated since the meaning was already established and stored as acquired vocabulary. Furthermore, pseudo-verbs reading required more support from the articulatory system for phonological processing [49], involving frontal regions of the left perisylvian circuit, which are part of the dorsal stream [3, 11]. In the present study, pseudo-verbs lacked meaning but simulated the meaning of verbs, perhaps because they were built with the expected ending of a verb in Spanish in the infinitive (i.e., *ar*, *er*, *ir*), thus providing functional morphology and requiring meaning processing [49, 50]. In sum, pseudo-verbs reading recruited the bilateral dorsolateral prefrontal cortex, involved in the search for meaning [13, 68], the superior lateral occipital cortex, including the supramarginal gyrus, a semantic hub [14, 21, 69], and modality-specific systems such as the visual word form area [70], as well as motor regions (somatosensory cortex, SMA, and right cerebellum [crus I, crus II, lobules VI, and VIIIa]). These findings suggest an integration for semantic representation between modality-general systems [71] and modality-specific systems, in conjunction with the core components of the language network. It is worth highlighting that, compared to verbs, reading pseudo-verbs did not recruit semantic brain areas from the temporal lobe. These findings confirm a robust interhemispheric dependency for the semantic representation of verbs as part of the ventral stream; and for pseudo-verbs, a higher involvement of the left language network, dorsal stream [3], including sensory-motor areas. Since these motor areas have been associated with action knowledge, phonological and articulatory processing, and executive functions such as verbal working memory [72], future studies should untangle whether the recruitment of motor-specific regions can be attributed to semantic grounding on sensory-motor processes or to another type of processing during pseudo-verbs reading.

The comparison between semantic categories of verbs, i.e., motor respect to mental verbs, yielded a null result, which supports the notion that motor verbs do not require additional neural mechanisms for semantic representation relative to abstract verbs [42]. However, it was surprising that when the effect of visual processing was removed from this comparison ([motor>symbols] > [mental>symbols], S1 Fig), motor verbs, being more concrete and readily imaginable, led to an increased BOLD signal in posterior brain regions associated with the representation of object knowledge. These posterior brain regions have been previously identified for noun processing [27, 28, 30] and include the superior lateral occipital cortex involved in storing meanings and visually identifying objects [21, 69] as well as the fusiform gyrus associated with the retrieval of visual object information and the strength of the behavioral

concreteness effect, concreteness refers to the degree to which a word alludes to features that can be sensually experienced [73].

On the other hand, as expected, mental verbs showed an increase in BOLD signal compared to motor verbs in the multimodal left language network (i.e., frontal and temporal regions) and in sensory-motor systems, including visual medial occipital regions, motor areas such as SMA, and the right cerebellum (crus I/II and lobule VI). It is worth noting that the activation observed in the left language network and cerebellum did not survive when the effect of symbols was removed, however, the medial frontal and occipital regions still exhibited an increase in BOLD signal. When the effect of pseudo-verbs was removed, mental verbs only recruited visual systems in the occipital regions, supporting the dominant role of the visual modality in sensory processing for the semantic representation of abstract concepts [44, 46]. Our mental verbs were classified as abstract and less imaginable and concrete than motor verbs, which could potentially lead to top-down modulation to enhance visual processing in the primary cortex for reading abstract verbs [74] and maybe use visual imagery or experiences to ground the semantic representation [9]. The primary visual regions with increased BOLD signal also showed functional connectivity with the right cerebellum, suggesting that this may be a network that supports the semantic representation of abstract mental verbs. Thus, consistent with the grounded cognition framework, our results suggest that abstract concepts related to mental action are grounded in modality-specific brain systems typically engaged in visual perception [21] that interact with multimodal systems for semantic representation [18, 75].

Left SMA and right cerebellum, in addition to the left-lateralized language network, showed an increased BOLD signal for both pseudo-verbs compared to verbs, and for mental verbs compared to motor verbs, while the primary motor cortex was only recruited for pseudo-verbs. Since the SMA and cerebellum are not primary associated with motor representations like the primary motor cortex, they might participate in a more abstract manner in the search for semantic representations. Thus, SMA and cerebellum could potentially anchor the abstract representations onto modality-specific sensorimotor mechanisms. This notion aligns with previous studies showing that the SMA, cerebellum, and medial posterior regions, including the precuneus, are nodes of the cognitive control network [76–78]. The SMA has been associated with task switching [79], response inhibition [80], and action selection and planning based on internal goals [13, 81, 82]. Simultaneously, the cerebellum is believed to create internal models of sensorimotor information for prediction and optimized execution [83–85].

As mentioned earlier, performance in verb fluency and lateralized motor activities (i.e., EHI scores) were found to be associated with the change in BOLD signal during the semantic contrast (mental vs. motor verbs) in two predefined semantic regions of interest obtained through an automated meta-analysis for 'verbs': the left superior lateral occipital cortex and the left middle temporal gyrus. The other two regions included were motor-related; the left supplementary motor area (SMA) and the right cerebellum. Firstly, we obtained functional connectivity for each semantic category of verbs and pseudo-verbs with respect to symbols. The left superior lateral occipital cortex, as a seed, exhibited functional connectivity with bilateral frontal, parietal, and temporal regions for both categories of verbs. However, for pseudo-verbs, it was only coupled with the left ventral frontal region, including primary sensorimotor regions. Studies have suggested that the left superior lateral occipital cortex functions as a semantic convergence zone [68, 86] that manages the meaning of stimuli. In the case of verbs, the left superior lateral occipital cortex was linked to regions that store conceptual knowledge, which are distributed widely. Conversely, for pseudo-verbs, this region and the left temporal middle gyrus showed functional connectivity with frontal regions involved in searching for meaning and learning [13, 87], somatosensory representations from the pre and postcentral gyri [87], articulatory and motor processing [11].

Interestingly, the motor seeds (left SMA, and right cerebellum) showed functional connectivity only for mental verbs compared to symbols. Left SMA was coupled with anterior regions, while right cerebellum with bilateral posterior regions such as temporal, medial occipital, sensorimotor regions, and polar frontal. These findings strengthen the idea that SMA and cerebellum collaborate to ground the abstract to modality-specific representations and support previous studies which have reported that conceptual meaning at the abstract level is embodied by the interaction of sensory and motor mechanisms with multimodal processing [75]. This interaction collects a variety of sparse bodily experiences [43, 52] or strategies [9, 11] for semantic representation.

Secondly, we computed functional connectivity for each category of verbs in comparison to pseudo-verbs using these four seeds. Among them, only three seeds demonstrated functional connectivity: the left superior lateral occipital cortex, left SMA, and right cerebellum. Once again, the superior lateral occipital cortex exhibited functional connectivity with an extended bilateral temporal and parietal network. Additionally, the left SMA demonstrated functional connectivity, for both motor and mental verbs, with bilateral superior temporal regions, the posterior cingulate cortex, and the frontal pole. Similarly, the right cerebellum displayed functional connectivity with other regions, but only for mental verbs, including bilateral temporal regions, the post and precentral gyri, frontal areas, and the precuneus. Finally, the connectivity analysis revealed that a portion (8.2%) of medial visual areas recruited by reading mental verbs compared to motor verbs were functionally coupled with the right cerebellum in the same semantic contrast, i.e., mental verbs versus motor verbs.

Our seed-ROI in the right cerebellum was located in crus I and lobule VI. The involvement of the cerebellum in language functions is strongly lateralized to the right hemisphere [15–17]. Specifically, lobules VI, VII, and crus I/II have shown functional connectivity with the cortical language network and support for semantic prediction in speech production and comprehension [88, 89] beyond its role in articulation as a motor component of language [78, 84]. The cerebellum is believed to create internal models of sensory-motor information and update them based on the comparison with the actual outcome [83–85]. Thus, during the reading of pseudo-verbs and mental compared to motor verbs, the cerebellum may have provided internal models of sensory-motor information to ground the representations or mapped the semantic representations when the referent was lacking [22]. Likewise, the right cerebellum demonstrated functional connectivity with the visual cortex when reading mental verbs, supporting the idea that the cerebellum creates sensory-motor internal models connected with the visual sensory system to optimize language comprehension [83–85, 90]. A recent study has shown that crus II and lobule VI in cerebellum are spontaneously and functionally coupled to the primary visual cortex [91] reports have involved both cerebellar areas with predictive coding on visual areas [74, 88, 89]. Our results strengthen the idea that the cerebellum is responsive to semantic information and is part of the language network that extends beyond the left language network. However, further studies are needed to understand the modulation exerted from the cerebellum to cortical areas during language processing. Exploring how the cerebellum modulates the flow of information between modality-specific systems in cortical areas will help explain its more significant involvement in abstract versus concrete concepts.

The current study's limitations are that other lexical levels, such as nouns or adjectives, were not included. Although the pseudo-verbs were constructed using Spanish phonological rules, their characteristics such as concreteness-abstraction, imageability, or arousal were not evaluated, which could be very useful for comparing them to semantic categories of verbs. Since it has been reported that different types of abstract concepts are associated with representational differences [18, 19], adding different categories of abstract verbs could have been useful.

In conclusion, this study confirmed differential activation and functional connectivity for reading verbs and pseudo-verbs. According to the dual stream model, the left dorsal stream that supports the sub-lexical route, and the extrasylvian motor system, including the right cerebellum, were involved in reading pseudo-verbs; while the ventral stream that maps words onto lexical conceptual representations was involved in reading verbs. Our findings support the embodied or grounded cognition model, indicating that modality-specific brain regions contribute to the semantic representation of abstract verbs in conjunction with the well-established multimodal left perisylvian networks. Additionally, we identified a preferential modality-specific system: visual systems were recruited by abstract verbs and exhibited functional connectivity with the right cerebellum, forming a network that supports the semantic representation of abstract concepts. These results confirm the dissociation between sub-lexical and lexical-semantic processing and provide evidence for the neurobiological basis of semantic representations grounded in modality-specific systems for abstract concepts.

## Supporting information

**S1 Fig. Comparison between motor verbs and mental verbs without the effect of symbols processing.** Graphical representation of GLM's results, brain regions activated in the contrast [motor > symbols] > [mental > symbols] in red, and [mental > symbol] > [motor>symbols] in green. Colorbars show z scores. LH: left hemisphere; RH: right hemisphere.
(TIF)

**S1 Table. Brain areas exhibiting significant activation in whole brain analysis during verbs > symbols and pseudo-verbs > symbols contrasts, according to GLM analysis.** The x, y, and z coordinates are in MNI space, regions were labelled according to Harvard-Oxford Cortical and Subcortical Atlases in FSLVIEW. L = Left region or hemisphere. R = Right region or hemisphere.
(PDF)

**S2 Table. Brain areas exhibiting significant activation in whole brain analysis during verbs > pseudo verbs and pseudo-verbs > verbs contrasts according to GLM analysis.** The x, y, and z coordinates are in MNI space, regions were labelled according to Harvard-Oxford Cortical and Subcortical Atlases in FSLVIEW. L = Left region or hemisphere. R = Right region or hemisphere.
(PDF)

**S3 Table. Brain areas exhibiting significant activation in whole brain analysis during mental > motor verbs, [mental > symbols] > [motor > symbols] and [mental > pseudo verbs] > [motor > pseudo verbs] contrasts, according to GLM analysis.** The x, y, and z coordinates are in MNI space, regions were labelled according to Harvard-Oxford Cortical and Subcortical Atlases in FSLVIEW. L = Left region or hemisphere. R = Right region or hemisphere.
(PDF)

**S4 Table. Brain areas exhibiting significant activation in whole brain analysis during [motor > symbols] > [mental > symbols] and [mental > symbols] > [motor > symbols] contrasts, according to GLM analysis.** The x, y, and z coordinates are in MNI space, regions were labelled according to Harvard-Oxford Cortical and Subcortical Atlases in FSLVIEW. L = Left region or hemisphere. R = Right region or hemisphere.
(PDF)

**S5 Table. Correlation between BOLD signal and behavioral performance.** L-SMG = left supramarginal gyrus (-60, -26, 28), R-STG = right superior temporal gyrus (56, -32, 2), L-LOC = left superior lateral occipital cortex (-26, -60, 48), L-MTG = left posterior middle temporal gyrus (-58, -10, -14). EHI = Edinburgh Handedness Inventory index.
(PDF)

**S6 Table. Brain areas exhibiting significant connectivity during motor > symbols, mental > symbols and peudo verbs > symbols contrasts, according to PPI analysis with seeds in L-LOC, L-MTG, L-SMA and R cerebellum.** The x, y, and z coordinates are in MNI space, regions were labelled according to Harvard-Oxford Cortical and Subcortical Atlases in FSLVIEW. L = Left region or hemisphere. R = Right region or hemisphere.
(PDF)

**S7 Table. Brain areas exhibiting significant connectivity during motor > pseudo verbs and mental > pseudo verbs, according to PPI analysis with seeds in L-LOC, L-SMA and R cerebellum.** The x, y, and z coordinates are in MNI space, regions were labelled according to Harvard-Oxford Cortical and Subcortical Atlases in FSLVIEW. L = Left region or hemisphere. R = Right region or hemisphere.
(PDF)

**S8 Table. Brain areas exhibiting significant connectivity during mental > motor verbs according to PPI analysis with seeds in right cerebellum.** The x, y, and z coordinates are in MNI space, regions were labelled according to Harvard-Oxford Cortical and Subcortical Atlases in FSLVIEW. L = Left region or hemisphere. R = Right region or hemisphere.
(PDF)

## Acknowledgments

We thank María del Carmen Barrios Giordano for editing the text.

## Author Contributions

**Conceptualization:** Azalea Reyes-Aguilar.

**Data curation:** Giovanna Licea-Haquet, Brenda I. Arce.

**Formal analysis:** Azalea Reyes-Aguilar.

**Funding acquisition:** Magda Giordano.

**Investigation:** Azalea Reyes-Aguilar, Giovanna Licea-Haquet, Brenda I. Arce, Magda Giordano.

**Methodology:** Azalea Reyes-Aguilar, Giovanna Licea-Haquet, Magda Giordano.

**Project administration:** Azalea Reyes-Aguilar, Magda Giordano.

**Resources:** Magda Giordano.

**Supervision:** Magda Giordano.

**Visualization:** Azalea Reyes-Aguilar, Giovanna Licea-Haquet.

**Writing – original draft:** Azalea Reyes-Aguilar, Giovanna Licea-Haquet.

**Writing – review & editing:** Azalea Reyes-Aguilar, Giovanna Licea-Haquet, Magda Giordano.

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
