## [Decision Letter · Decision Letter 0]

22 Jun 2023

PONE-D-23-14317Contribution and functional connectivity between cerebrum and cerebellum on sub-lexical and lexical-semantic processing of verbsPLOS ONE

Dear Dr. Reyes-Aguilar,

Thank you for submitting your manuscript to PLOS ONE. After careful consideration, we feel that it has merit but does not fully meet PLOS ONE’s publication criteria as it currently stands. Therefore, we invite you to submit a revised version of the manuscript that addresses the points raised during the review process.

We look forward to receiving your revised manuscript.

Kind regards,

Florian Ph.S Fischmeister

Academic Editor

PLOS ONE

“MG: This study was supported by grants from DGAPA-PAPIIT IN203818 and

CONACyT Fronteras de la Ciencia #225.”

Reviewers' comments:

Reviewer's Responses to Questions

**Comments to the Author**

1. Is the manuscript technically sound, and do the data support the conclusions?

Reviewer #1: Partly

Reviewer #2: Partly

2. Has the statistical analysis been performed appropriately and rigorously? 

Reviewer #1: Yes

Reviewer #2: No

3. Have the authors made all data underlying the findings in their manuscript fully available?

Reviewer #1: Yes

Reviewer #2: No

4. Is the manuscript presented in an intelligible fashion and written in standard English?

Reviewer #1: Yes

Reviewer #2: No

5. Review Comments to the Author

Reviewer #1: The research question is clearly presented, the stimuli set has been built rigorously, data analyses have been conducted appropriately.

Detailed comments:

- the neuroanatomical correlates of the N/V distinction are much more controversial than they are referred to in 94-95;

- 96-97: 1) word class should be used here, rather than "lexical" class; 2) the Authors might acknowledge that the verb category comprises a wide range of functional properties that cannot be reduced to the dichotomy between concrete motion vs. abstract mental actions; 3) [+abstract] verbs include states (such as prefer, know, etc.) that cannot be subsumed under the category of mental actions as defined here. Point 3 deserves further attention and clarification;

- the relationship between visual components and [+abstract] (non-motor) actions, such as 'understand', should be better clarified; this is particularly relevant, as it is related to both the expected outcome and the result on the involvement of the visual modality-specific regions in processing mental verbs;

- the interpretation of the following result (see 522-524) needs further clarification: "verb reading did not recruit the dorsolateral prefrontal cortex since meaning selection is already established and stored in the mental lexicon";

- the interpretation of the null result from the contrast mental vs. motor verbs needs further clarification. The claim in 543-544 should be better explained: the relationship between concreteness hierarchy and the activation of neural mechanisms for semantic representation should be discussed in relation to further evidence;

- the differential activation and functional connectivity for reading verbs vs. pseudo-verbs should be discussed in relation to neuroanatomical and functional evidence on processing words vs. pseudo-words (see previous studies by E. Fedorenko and collaborators: e.g., 2010, 2012, etc.);

- reference num. 30 is incomplete.

Reviewer #2: The authors present a manuscript investigating functional connectivity of sub-lexical and lexical-semantic processing of verbs using fMRI.

Although the topic is highly interesting, there are some major issues regarding the presented work:

1) Handedness was assessed for all participants using EHI, however there is no information on handedness of included participants. The authors only presented a mean and correlations. I would suggest to include only right-handed participants, as previous research has shown that language processing of people with left dominant hand may be less lateralized compared to people with right hand dominance.

2) Writing: The description of the task is very complex and partly confusing. I would therefore suggest to review this part. Moreover, the hypothesis should be placed at the end of the introduction, and should not be part of the methods section.

3) Statistics: The authors present data corrected for multiple testing as well as uncorrected results (lack of statistical significance?) The results should be presented with correction for multiple testing (same procedure for all tests), the procedure and the significance level should be part of the methods section. In addition, please report exact p-values and effect sizes.

4) Verbal fluency task: Why did the authors chose the letter M? Many other studies (mostly English) used letter F,A or S.

Previous research already reported poorer result of phonemic fluency compared to semantic fluency. Please cite respective research.

6. PLOS authors have the option to publish the peer review history of their article (what does this mean?). If published, this will include your full peer review and any attached files.

Reviewer #1: No

Reviewer #2: **Yes: **Kathrin Kollndorfer

---

## [Author Response · Author response to Decision Letter 0]

5 Aug 2023

Florian Ph. S Fischmeister,

We sincerely appreciate your provision of additional requirements for the revision of our manuscript. We have thoroughly reviewed the points raised and would like to address them as follows:

Style Requirements: We apologize for any oversight in adhering to PLOS ONE's style requirements, including file naming. We have carefully examined the PLOS ONE style templates and will ensure that our revised manuscript strictly adheres to these style guidelines, including appropriate file naming.

Role of Funders: We are grateful for your acknowledgment of the financial disclosure provided. We confirm that the funders, DGAPA-PAPIIT IN203818 and CONACyT Fronteras de la Ciencia #225, had no involvement in the study design, data collection and analysis, decision to publish, or preparation of the manuscript. As a result, we have included the following statement in our cover letter: "The funders had no role in the study design, data collection and analysis, decision to publish, or preparation of the manuscript."

Data Sharing: We apologize for any misunderstanding regarding the reference to "data not shown" in our manuscript. We have now provided the appropriate URLs (https://openneuro.org/datasets/ds003481/versions/1.0.3/download) for the open data repository and the DOI (https://doi.org/10.3389/fnhum.2021.666210) where the data description can be found in a publication.

Supporting Information: Thank you for bringing to our attention the need to include captions for our Supporting Information files at the end of the manuscript. We have included the required captions for all Supporting Information files and have updated any in-text citations accordingly. The changes have been highlighted in yellow.

We greatly value your guidance in ensuring that our manuscript aligns with the requirements of PLOS ONE. We have made the necessary revisions as outlined above.

Dear Reviewer #1:

Thank you for your valuable feedback and insightful comments on our manuscript. Your suggestions greatly enhanced the clarity and quality of our manuscript. We have carefully considered each of your detailed comments and addressed them accordingly:

Reviewer #1: -the neuroanatomical correlates of the N/V distinction are much more controversial than they are referred to in 94-95;

Answer: Regarding the neuroanatomical correlates of the Nouns/Verbs distinction, we acknowledge that the topic is indeed controversial. We revised the related statements in the manuscript to reflect the existing debates more accurately. 

pp. 5

It now reads (lines 92 to 102):

Although most studies on semantic representation have focused on nouns, verbs have become increasingly important in studying both semantic and syntactic processing. Noun processing has been associated with object knowledge representation in the left inferior temporal regions, while verb processing has been considered more complex, involving action knowledge, morphosyntactic processing, and executive functions related to the prefrontal and motor cortex [27–31,32,33]. There is demonstrated involvement of motor and pre-motor brain regions during the reading of verbs denoting motor actions [26,34,35]. However, when semantic content is matched between nouns and verbs, such as comparing action nouns to action verbs, the differences between nouns and verbs become minimal or nonexistent, challenging psycholinguistic models that support the organizational principle of grammatical class in lexical knowledge [36–38].

Reviewer #1: - 96-97: 1) word class should be used here, rather than "lexical" class; 2) the Authors might acknowledge that the verb category comprises a wide range of functional properties that cannot be reduced to the dichotomy between concrete motion vs. abstract mental actions; 3) [+abstract] verbs include states (such as prefer, know, etc.) that cannot be subsumed under the category of mental actions as defined here. Point 3 deserves further attention and clarification;

Answer: We appreciate your suggestions regarding the categorization of verbs. We incorporated the use of "word class" instead of "lexical class" in the appropriate sections. Additionally, we recognized the need to acknowledge the wide range of functional properties within the verb category beyond the concrete motion vs. abstract mental actions dichotomy. We further clarified this point and provide a more comprehensive discussion of [+abstract] verbs, including states, to avoid oversimplification.

pp. 5-6

It now reads (lines 102-115):

Verbs, as a fundamental word class, typically represent actions belonging to various semantic classes, including mental, relational, motor, verbal, existence, modulation, and states [39]. Meaning is multidimensional, encompassing various experiential components or aspects that can be encoded or evoked simultaneously, which allows for the possibility of a single verb being associated with distinct semantic classes [39]. For example, the verb "turn" can be associated with the motor semantic class, indicating the action of rotating or changing the direction of an object. It can also be linked to the state semantic class, indicating the process of becoming something or undergoing a change in nature or character. However, in this study, our specific focus is on two semantic classes of verbs: concrete-motor movement verbs, which encompass various subclasses of physical (non-mental) processes related to movements in space, posture-position, and modification of objects (e.g., 'write'), and abstract-mental action verbs (e.g., 'think') which were selected as they refer to entities endowed with psychic life that maintain or experience different states, changes of state, or inner perceptual, sensory, and/or cognitive activities [39–41].

Reviewer #1: - the relationship between visual components and [+abstract] (non-motor) actions, such as 'understand', should be better clarified; this is particularly relevant, as it is related to both the expected outcome and the result on the involvement of the visual modality-specific regions in processing mental verbs.

Answer: We understand the importance of clarifying the relationship between visual components and [+abstract] (non-motor) actions, such as 'understand, particularly concerning its relevance to outcome and the involvement of visual modality-specific regions in processing mental verbs. We have addressed this point as suggested.

pp. 6-7

It now reads (lines 133-143): 

Sensory and motor information has been found to play a significant role in the semantic content of abstract nouns [9,18–21]. Harpaintner et al., (2020) found that abstract concepts related to action and vision are grounded in modality-specific brain systems typically engaged in actual perception and action. In the case of verbs, another study [44] identified examples of abstract verbs that elicit similar outcomes to visual imagery when they are straightforwardly interpreted as referring to spatially concrete scenes. In this regard, visual brain regions have been shown to have a significant role in processing certain abstract verbs (mental verbs such as 'understand') that are associated with visual information, such as “see” or “look” [19,22]. Vision often plays a dominant role in our interpretation of sensory information [45] and extends to the semantics of abstract concepts, with hearing a close second [46].

Reviewer #1: - the interpretation of the following result (see 522-524) needs further clarification: "verb reading did not recruit the dorsolateral prefrontal cortex since meaning selection is already established and stored in the mental lexicon";

Answer: We understand the need for further clarification in the interpretation of the result regarding the dorsolateral prefrontal cortex and its role in verb reading. We have provided a more detailed explanation.

pp. 24

Now it reads (lines 561-572):

In keeping with our expectations, verbs—in contrast to symbols and pseudo-verbs—recruited an extended reading network on bilateral temporal regions that belong to the ventral stream, related to semantic processing [1,3] and medial regions related to attention [8,68]. On the other hand, pseudo-verbs compared to verbs recruited prefrontal regions, perhaps related to Hagoort’s (2015) proposition that language processing recruits various prefrontal cortex regions: the ventral region integrates elements from memory to create novel meanings or interpretations, while the dorsolateral region, associated with attentional control, interacts with the language network to select the most relevant information for language comprehension. In the present study, the reading of pseudo-verbs recruited the prefrontal cortex possibly in a search for meaning for these unknown linguistic stimuli. In contrast, in the case of verbs, it seems there would be no need for the prefrontal regions to be selectively activated since the meaning was already established and stored as acquired vocabulary.

Reviewer #1: - the interpretation of the null result from the contrast mental vs. motor verbs needs further clarification. The claim in 543-544 should be better explained: the relationship between concreteness hierarchy and the activation of neural mechanisms for semantic representation should be discussed in relation to further evidence.

Answer: The interpretation of the null result from the contrast between mental and motor verbs has been expanded and clarified in the revised manuscript. We addressed the relationship between the concreteness hierarchy and the activation of neural mechanisms for semantic representation considering additional relevant evidence.

pp. 25

Now it reads (lines 595-606):

The comparison between semantic categories of verbs, i.e., motor respect to mental verbs, yielded a null result, which supports the notion that motor verbs do not require additional neural mechanisms for semantic representation relative to abstract verbs [42]. However, it was surprising that when the effect of visual processing was removed from this comparison ([motor>symbols] > [mental>symbols], S1 Fig), motor verbs, being more concrete and readily imaginable, led to an increased BOLD signal in posterior brain regions associated with the representation of object knowledge. These posterior brain regions have been previously identified for noun processing [27,28,30] and include the superior lateral occipital cortex involved in storing meanings and visually identifying objects [21,70] as well as the fusiform gyrus associated with the retrieval of visual object information and the strength of the behavioral concreteness effect, concreteness refers to the degree to which a word alludes to features that can be sensually experienced [74].

Reviewer #1: - the differential activation and functional connectivity for reading verbs vs. pseudo-verbs should be discussed in relation to neuroanatomical and functional evidence on processing words vs. pseudo-words (see previous studies by E. Fedorenko and collaborators: e.g., 2010, 2012, etc.);

Answer: The differential activation and functional connectivity for reading verbs vs. pseudo-verbs has been discussed in relation to previous papers by Fedorenko and collaborators. Additionally, in introduction was included the findings of this research group.

pp. 24-25

It now reads (lines 573 to 585):

Furthermore, pseudo-verbs reading required more support from the articulatory system for phonological processing [49], involving frontal regions of the left perisylvian circuit, which are part of the dorsal stream [3,11]. In the present study, pseudo-verbs lacked meaning but simulated the meaning of verbs, perhaps because they were built with the expected ending of a verb in Spanish in the infinitive (i.e., ar, er, ir), thus providing functional morphology and requiring meaning processing [49,50]. In sum, pseudo-verbs reading recruited the bilateral dorsolateral prefrontal cortex, involved in the search for meaning [13,68], the superior lateral occipital cortex, including the supramarginal gyrus, a semantic hub [21,69,70], and modality-specific systems such as the visual word form area [71], as well as motor regions (somatosensory cortex, SMA, and right cerebellum [crus I, crus II, lobules VI, and VIIIa]). These findings suggest an integration for semantic representation between modality-general systems [72] and modality-specific systems, in conjunction with the core components of the language network.

Reviewer #1: - reference num. 30 is incomplete.

Answer: We apologize for the incomplete reference (num. 30, now 42) and will ensure its completeness in the revised manuscript.

Dear Reviewer #2:

Thank you for your valuable feedback on our manuscript investigating the functional connectivity of sub-lexical and lexical-semantic processing of verbs using fMRI. We appreciate your insightful comments and suggestions. We would like to inform you that we have addressed your concerns and made the necessary revisions to improve the clarity and scientific rigor of our study.

Reviewer #2: 1) Handedness was assessed for all participants using EHI, however there is no information on handedness of included participants. The authors only presented a mean and correlations. I would suggest to include only right-handed participants, as previous research has shown that language processing of people with left dominant hand may be less lateralized compared to people with right hand dominance.

Answer: We apologize for the oversight in not providing information on the handedness of the participants in the initial version of the manuscript. We have now included this information in the revised manuscript. Now in Methods/Behavioral testing section, pp. 12, lines 270 to 274, it reads: 

Finally, the Edinburgh Handedness Inventory (EHI [40]) was administered to assess hand dominance in everyday motor activities. The EHI index, which ranges from -100 to 100, was calculated. This index enables the classification of participants with negative scores from -40 to -100 as left-handed, scores between -40 and 40 as ambidextrous, and scores greater than 40 as right-handed.

And, in the Results section, pp. 17, lines 388 to 392: 

According to the EHI scores (M = 76.04, SD = 21.37, range = 10-100), which serve as a measure of everyday motor behavior laterality, 8.84% of participants were classified as ambidextrous (score < 40), while 91.67% were right-handed (score >= 40). No left-handed individuals were included in the study. Finally, we explored the relationship between EHI scores and performance on the language tasks: none of the results remained statistically significant after applying multiple comparison correction (FDR).

As you mentioned, previous studies by Nathalie Tzourio-Mazoyer have suggested that individuals with left-hand dominance may show reduced language lateralization compared to individuals with right or mixed hand dominance (Mazoyer et al., 2014; Zago et al., 2016). Therefore, left-handed individuals were not included in this study to maintain consistency with previous research findings on language processing and hand dominance.

- Mazoyer, B., Zago, L., Jobard, G., Crivello, F., Joliot, M., Perchey, G., Mellet, E., Petit, L., Tzourio-Mazoyer, N. (2014). Gaussian mixture modeling of hemispheric lateralization for language in a large sample of healthy individuals balanced for handedness. PloS one, 9(6), e101165.

- Zago, L., Petit, L., Mellet, E., Jobard, G., Crivello, F., Joliot, M., et al. (2016). The association between hemispheric specialization for language production and for spatial attention depends on left-hand preference strength. Neuropsychologia 93, 394–406. doi: 10.1016/j.neuropsychologia.2015.11.018

Reviewer #2: 2) Writing: The description of the task is very complex and partly confusing. I would therefore suggest to review this part. Moreover, the hypothesis should be placed at the end of the introduction, and should not be part of the methods section.

Answer: We appreciate your feedback regarding the description of the task. We have revised this section to improve clarity and ensure that it is more easily understandable for readers. Now that section reads in pp. 10-11, lines 233 to 245):

We included four stimulus categories: mental verbs, motor verbs, pseudo-verbs, and symbols. The stimuli were presented using a block design, each block consisting of stimuli from the same category. There were seven different stimuli in each block, three of which were repeated consecutively, resulting in a total of ten stimuli per block. The stimuli were displayed for 1000 ms with interstimulus intervals of 2000 ms. Blank intervals of 12 seconds separated the blocks (Fig 1). Two blocks of each stimulus category were presented pseudo-randomly during each run. The total number of runs was eight. Each run lasted approximately 6 minutes. Participants were instructed to identify the current stimulus (i.e., verbs, pseudo-verbs, and symbols strings) and indicate during the stimulus presentation whether it was the same as the previous one (one-back detection task). The stimuli were presented on a gray background using the PsychoPy® software [56,57], and a projection system consisting of MR-compatible goggles from NordicNeuroLab (Bergen, Norway). Additionally, a button system from NordicNeuroLab was used to record the participants' responses.

Despite the task being designed in blocks, the analysis was conducted by modeling the activity related to each stimulus, similar to an event-related design. To accomplish this, a regressor was created to represent the duration of stimulus presentation (1000 ms) for each block's ten stimuli. The above was clarified in the section describing the fMRI data analysis as follows in pp. 13, lines 299 to 301:

The regressors were created to specifically represent the duration of stimulus presentation (1000 ms) for each set of ten stimuli within the block, resembling an event-related design.

Additionally, we have moved the hypothesis to the end of the introduction, as per your suggestion, to enhance the logical flow of the manuscript. It now reads, pp.7-8, lines 151 to 168:

To compare the contribution and functional connectivity of brain regions in sub-lexical and lexical-semantic processing of verbs, we designed stimuli that encompassed pseudo-verbs, strings of letters featuring characteristic verb infinitive endings in Spanish (i.e., ar, er, ir), and actual verbs. While reading or identifying stimuli engage similar visual processing flows regardless of stimulus type, only verbs and pseudo-verbs require phonological processing; similarly, only verbs involve lexical-semantic information [4,49,50]. We consequently contrasted verbs, which we expected would recruit the ventral stream, with pseudo-verbs, which would activate the dorsal stream. Likewise, we were interested in the dichotomous brain representation of concrete versus abstract verbs because, as mentioned above, the brain correlates are controversial [18,23,26,51]. Thus, we were evaluating the involvement of integrating multimodal and modality-specific brain regions in the processing of verb categories. According to the embodied or grounded cognition approach, we would expect to find that both multimodal and modality-specific cortical regions would be involved in processing both categories [18,52]. However, we expected a variation in involvement of modality-specific regions between verb categories: motor verbs would primarily engage motor brain areas, while mental verbs would activate sensory brain areas, with a dominant focus on visual processing, as observed in previous studies [44,46].

Reviewer #2: 3) Statistics: The authors present data corrected for multiple testing as well as uncorrected results (lack of statistical significance?) The results should be presented with correction for multiple testing (same procedure for all tests), the procedure and the significance level should be part of the methods section. In addition, please report exact p-values and effect sizes.

Answer: We appreciate your comment regarding the presentation of the statistical results. In the revised manuscript, we have provided corrections for multiple testing for all analyses, ensuring a consistent procedure, Fig 2 was modified to be consistent with these changes, as were some parts of discussion. The methods section now includes detailed information on the statistical procedures used, including the significance level. We have also included exact p-values and effect sizes to provide a comprehensive understanding of the results.

In Methods, pp.16, lines 368 to 372: 

Behavioral data were analyzed using R 3.6.1. We compared the performance among three verbal fluency tasks using a repeated-measures ANOVA, and calculated Pearson’s correlation coefficient to test whether the scores of the behavioral tasks were correlated. We also set a false discovery rate (FDR) correction to determine the threshold for significant differences (p<0.05).

In Results, pp. 16-17, lines 376 to 395: 

We observed a higher fluency of verbs (M = 24.50, SD = 6.87, range = 8-36, F (2, 46) = 33.66, η2 = 0.37, p < 0.0000000218, FDR corrected) and semantic fluency (M = 23.95, SD = 4.08, range = 15-33, p < 0.000000398, FDR corrected) relative to phonological fluency (M = 16.17, SD = 3.86, range = 8-26). No significant difference was found between semantic and verb fluency. We then examined the relationship between measures of verbal fluency: only verb fluency demonstrated a positive correlation with phonological fluency (r = 0.55, p = 0.014, Pearson, FDR corrected). Next, we evaluated the relationship between verbal fluency and the vocabulary task (M = 30.67, SD = 8.78, range = 8-43). The vocabulary task showed a positive correlation with both verb fluency (r = 0.60, p = 0.006, Pearson, FDR corrected) and semantic fluency (r = 0.54, p = 0.009, Pearson, FDR corrected, Fig 2), but we observed no significant correlation with phonological fluency. According to the EHI scores (M = 76.04, SD = 21.37, range = 10-100), which serve as a measure of everyday motor behavior laterality, 8.84% of participants were classified as ambidextrous (score < 40), while 91.67% were right-handed (score >= 40). No left-handed individuals were included in the study. Finally, we explored the relationship between EHI scores and performance on the language tasks: none of the results remained statistically significant after applying multiple comparison correction (FDR).

Fig 2. Correlations among scores on behavioral tasks. Vocabulary task of WAIS-IV, verb, semantic, and phonological fluency (f).

In Results, pp. 20; lines 464 to 477: 

The BOLD signal change for mental > motor verbs contrast in the L-LOC ROI was positively correlated to verbs fluency (r = 0.53, p = 0.034, Pearson, FDR corrected) and negatively correlated to EHI (r = -0.50, p = 0.034, Pearson, FDR corrected). The L-MTG ROI was negatively correlated to EHI (r = -0.54, p = 0.030, Pearson, FDR corrected). Other significant correlations were detected between behavioral measures and BOLD signal change for other contrasts: verbs > symbols and verbs > pseudo-verbs (S5 Table). Since a significant correlation was observed between the BOLD signal in the mental > motor contrast with verbs fluency and EHI, we conducted an exploratory analysis to calculate partial correlations between EHI and BOLD signal, with verbs fluency as the covariate. For this, negative partial correlations were found for L-MTG (r = -0.58, p = 0.003, Pearson), both motor areas, L-SMA (r = -0.45, p = 0.029, Pearson), and R-cerebellum (r = -0.42, p = 0.045, Pearson). Therefore, we identified these four ROIs—L-LOC and L-MTG as semantic regions, and L-SMA and R-Cerebellum as motor areas—as having a significant relationship with verb fluency or EHI, making them suitable candidates for the analysis of functional connectivity.

Reviewer #2: 4) Verbal fluency task: Why did the authors chose the letter M? Many other studies (mostly English) used letter F,A or S.

Previous research already reported poorer result of phonemic fluency compared to semantic fluency. Please cite respective research.

Answer: We chose the letter M in Spanish because of its frequency of use which is 3.43% according to the Diccionario del Español en México (Dictionary of Spanish in Mexico), comparable to the mean frequency of letters used in previous studies. For instance, a cross-cultural study with English and Spanish speakers used the letters C, F, and L for English, and P, S and V for Spanish speakers (Jacobs et al., 1997). In both languages the mean frequency of use of those letters is 3% (Dicionario del Español en México; Agyepong et al., 2018). Another study with Italian speakers used the letters F, A and S, with a mean frequency of 5.78% (Costa et al., 2013; https://www.sttmedia.com/wordcreator), while a study with Dutch speakers used the letter M which has a frequency of 2.6% (Shao et al., 2014; https://www.sttmedia.com/wordcreator). The use of the letter M made the task moderately challenging for our young and healthy participants, allowing us to ascertain that they were fluent in their native tongue and similar among them.

We have added the relevant information to the manuscript in pp.12, lines 266 to 269, now reads:

The letter “m” in Spanish has a frequency of use of 3.4% (Diccionario del Español de México, http://dem.colmex.mx), making the task moderately challenging for the participants and within the range of use frequency (from 1 to 10%) of letters selected in previous studies (e.g., 59–61).

Regarding differences in phonemic versus semantic fluency, indeed it is a common finding that neurotypical adults show poorer performance in phonemic (letter) fluency than semantic (category) fluency. We have cited a sample of studies that exemplify this finding. 

pp. 23, lines 549 to 551, it now reads:

The results of the behavioral tests showed better linguistic performance for verbs and semantic fluency than phonological fluency, consistent with previous findings among Spanish speakers and speakers of other languages [59–61,66,67].

For both reviewers:

Once again, we sincerely appreciate your thorough review and valuable suggestions. We believe that the revisions we have implemented have enhanced the manuscript, and we trust that you will find the updated version to be more satisfactory.

Furthermore, we would like to notify you that as part of the revision process, we have refined the wording of the text. We extend our gratitude to María del Carmen Barrios Giordano for her diligent editing.

Lastly, we have included the following list of references in the manuscript as part of this review:

• Damasio AR, Tranel D. Nouns and verbs are retrieved with differently distributed neural systems. Vol. 90, Proc. Natl. Acad. Sci. USA. 1993.

• Daniele A, Giustolisi L, Silveri MC, Colosimo C, Gainottit G. Evidence for a possible neuroanatomical basis for lexical processing of nouns and verbs. 1994;32(11):1325–41.

• Tyler LK, Bright P, Fletcher P, Stamatakis EA. Neural processing of nouns and verbs: The role of inflectional morphology. Neuropsychologia. 2004;42(4):512–23.

• Shapiro KA, Moo LR, Caramazza A. Cortical signatures of noun and verb production [Internet]. Vol. 103. 2006. Available from: www.pnas.orgcgidoi10.1073pnas.0504142103

• Siri S, Tettamanti M, Cappa SF, Rosa P Della, Saccuman C, Scifo P, et al. The neural substrate of naming events: Effects of processing demands but not of grammatical class. Cerebral Cortex. 2008 Jan;18(1):171–7.

• Alyahya RSW, Halai AD, Conroy P, Lambon Ralph MA. Noun and verb processing in aphasia: Behavioural profiles and neural correlates. Neuroimage Clin. 2018;18:215–30.

• Vigliocco G, Vinson DP, Siri S. Semantic similarity and grammatical class in naming actions. Cognition. 2005;94(3).

• Saccuman MC, Cappa SF, Bates EA, Arevalo A, Della Rosa P, Danna M, et al. The impact of semantic reference on word class: An fMRI study of action and object naming. Neuroimage. 2006 Oct 1;32(4):1865–78.

• Albertuz F. Sintaxis, semántica y clases de verbos: Clasificación verbal en el proyecto ADESSE. In: Actas del VI Congreso de Lingüística General. Santiago de Compostela; 2007.

• García-Miguel JM, Costas L, Martínez S. Diátesis verbales y esquemas construccionales. Verbos, clases semánticas y esquemas sintáctico-semánticos en el proyecto ADESSE. In: Lang P, editor. Entre semántica léxica, teoría del léxico y sintaxis. Frankfurt : Wotjak, Gerd, & Juan Cuartero Otal; 2005. p. 373–84.

• García-Miguel JM, Francisco J. Albertuz. Verbs, Semantic Classes and Semantic Roles in the ADESSE project. In: Erk K, Melinger A, Schulte im Walde S, editors. Proceedings of the Interdisciplinary Workshop on the Identification and Representation of Verb Features and Verb Classes . Saarbrücken; 2005.

• Bergen BK, Lindsay S, Matlock T, Narayanan S. Spatial and linguistic aspects of visual imagery in sentence comprehension. Cogn Sci. 2007 Sep;31(5):733–64.

• San Roque L, Kendrick KH, Norcliffe E, Brown P, Defina R, Dingemanse M, et al. Vision verbs dominate in conversation across cultures, but the ranking of non-visual verbs varies. Cogn Linguist. 2015 Feb 1;26(1):31–60.

• Palmer SE. Vision science: Photons to phenomenology. Cambridge: Bradford Books.; 1999.

• Fedorenko E, Hsieh PJ, Nieto-Castañón A, Whitfield-Gabrieli S, Kanwisher N. New method for fMRI investigations of language: Defining ROIs functionally in individual subjects. J Neurophysiol. 2010 Aug;104(2):1177–94.

• Fedorenko E, Nieto-Castañón A, Kanwisher N. Syntactic processing in the human brain: What we know, what we don’t know, and a suggestion for how to proceed. Brain Lang. 2012 Feb;120(2):187–207. 

• Fedorenko E, Duncan J, Kanwisher N. Language-selective and domain-general regions lie side by side within Broca’s area. Current Biology. 2012 Nov 6;22(21):2059–62.

• Fedorenko E, Blank IA, Siegelman M, Mineroff Z. Lack of selectivity for syntax relative to word meanings throughout the language network. Cognition. 2020 Oct 1;203.

• Jacobs DM, Sano M, Albert S, Schofield P, Dooneief G, Stern Y. Cross-cultural neuropsychological assessment: A comparison of randomly selected, demographically matched cohorts of English- and Spanish-speaking older adults. J Clin Exp Neuropsychol. 1997;19(3):331–9.

• Costa A, Bagoj E, Monaco M, Zabberoni S, De Rosa S, Papantonio AM, et al. Standardization and normative data obtained in the Italian population for a new verbal fluency instrument, the phonemic/semantic alternate fluency test. Neurological Sciences. 2014 Mar;35(3):365–72. 

• Shao Z, Janse E, Visser K, Meyer AS. What do verbal fluency tasks measure? Predictors of verbal fluency performance in older adults. Front Psychol. 2014;5(JUL). 

• Vaughan RM, Coen RF, Kenny RA, Lawlor BA. Preservation of the semantic verbal fluency advantage in a large population-based sample: normative data from the TILDA study. Journal of the International Neuropsychological Society. 2016 May 1;22(5):570–6.

• Fernández KK, Kociolek AJ, Lao PJ, Stern Y, Manly JJ, Vonk JMJ. Longitudinal decline in semantic versus letter fluency, but not their ratio, marks incident Alzheimer’s disease in Latinx Spanish-speaking older individuals . Journal of the International Neuropsychological Society. 2023 Jan 13;1–8.

• Diccionario del Español en México (DEM). http://dem.colmex.mx. El Colegio de México. A.C., [consulted July 5th 2023]

• Willems RM, Hagoort P, Casasanto D. Body-specific representations of action verbs: Neural evidence from right- and left-handers. Psychol Sci. 2010;21(1):67–74.

• Zapała D, Iwanowicz P, Francuz P, Augustynowicz P. Handedness effects on motor imagery during kinesthetic and visual-motor conditions. Sci Rep. 2021 Dec 1;11(1).

• Fliessbach K, Weis S, Klaver P, Elger CE, Weber B. The effect of word concreteness on recognition memory. Neuroimage. 2006 Sep;32(3):1413–21.

---

## [Decision Letter · Decision Letter 1]

1 Sep 2023

Contribution and functional connectivity between cerebrum and cerebellum on sub-lexical and lexical-semantic processing of verbs

PONE-D-23-14317R1

Dear Dr. Reyes-Aguilar,

We’re pleased to inform you that your manuscript has been judged scientifically suitable for publication and will be formally accepted for publication once it meets all outstanding technical requirements.

Kind regards,

Florian Ph.S Fischmeister

Academic Editor

PLOS ONE

Additional Editor Comments (optional):

Reviewers' comments:

Reviewer's Responses to Questions

**Comments to the Author**

1. If the authors have adequately addressed your comments raised in a previous round of review and you feel that this manuscript is now acceptable for publication, you may indicate that here to bypass the “Comments to the Author” section, enter your conflict of interest statement in the “Confidential to Editor” section, and submit your "Accept" recommendation.

Reviewer #2: All comments have been addressed

Reviewer #3: (No Response)

2. Is the manuscript technically sound, and do the data support the conclusions?

Reviewer #2: Yes

Reviewer #3: Yes

3. Has the statistical analysis been performed appropriately and rigorously? 

Reviewer #2: Yes

Reviewer #3: Yes

4. Have the authors made all data underlying the findings in their manuscript fully available?

Reviewer #2: (No Response)

Reviewer #3: Yes

5. Is the manuscript presented in an intelligible fashion and written in standard English?

Reviewer #2: (No Response)

Reviewer #3: Yes

6. Review Comments to the Author

Reviewer #2: (No Response)

Reviewer #3: I was invited as a third reviewer after the submission of the first revision by the authors. I find this paper clear and well-written and ready to be published. In my opinion, the authors managed to incorporate all suggestions and critics.

7. PLOS authors have the option to publish the peer review history of their article (what does this mean?). If published, this will include your full peer review and any attached files.

Reviewer #2: No

Reviewer #3: No

---

## [Editor Report · Acceptance letter]

6 Sep 2023

PONE-D-23-14317R1 

Contribution and functional connectivity between cerebrum and cerebellum on sub-lexical and lexical-semantic processing of verbs 

Dear Dr. Reyes-Aguilar:

I'm pleased to inform you that your manuscript has been deemed suitable for publication in PLOS ONE. Congratulations! Your manuscript is now with our production department. 

Kind regards, 

on behalf of

Mag. Dr. Florian Ph.S Fischmeister 

Academic Editor

PLOS ONE